# Effect of Short-Term Restraint Stress on the Hypothalamic Transcriptome Profiles of Rats with Inherited Stress-Induced Arterial Hypertension (ISIAH) and Normotensive Wistar Albino Glaxo (WAG) Rats

**DOI:** 10.3390/ijms25126680

**Published:** 2024-06-18

**Authors:** Dmitry Yu. Oshchepkov, Yulia V. Makovka, Larisa A. Fedoseeva, Alisa A. Seryapina, Arcady L. Markel, Olga E. Redina

**Affiliations:** 1Federal Research Center Institute of Cytology and Genetics, Siberian Branch of Russian Academy of Sciences, 630090 Novosibirsk, Russia; diman@bionet.nsc.ru (D.Y.O.); makovkayv@bionet.nsc.ru (Y.V.M.); fedoseeva@bionet.nsc.ru (L.A.F.); seryapina@bionet.nsc.ru (A.A.S.); markel@bionet.nsc.ru (A.L.M.); 2Kurchatov Genomic Center Institute of Cytology and Genetics, Siberian Branch of Russian Academy of Sciences, 630090 Novosibirsk, Russia; 3Department of Natural Sciences, Novosibirsk State University, 630090 Novosibirsk, Russia

**Keywords:** hypothalamus, short-term restraint stress, gene expression, hypertension, RNA-Seq, ISIAH rat strain

## Abstract

Emotional stress is one of the health risk factors in the modern human lifestyle. Stress exposure can provoke the manifestation of various pathological conditions, one of which is a sharp increase in the blood pressure level. In the present study, we analyzed changes in the transcriptome profiles of the hypothalamus of hypertensive ISIAH and normotensive WAG rats exposed to a single short-term restraint stress (the rat was placed in a tight wire-mesh cage for 2 h). This type of stress can be considered emotional stress. The functional annotation of differentially expressed genes allowed us to identify the most significantly altered biological processes in the hypothalamus of hypertensive and normotensive rats. The study made it possible to identify a group of genes that describe a general response to stress, independent of the rat genotype, as well as a hypothalamic response to stress specific to each strain. The alternatively changing expression of the *Npas4* (neuronal PAS domain protein 4) gene, which is downregulated in the hypothalamus of the control WAG rats and induced in the hypothalamus of hypertensive ISIAH rats, is suggested to be the key event for understanding inter-strain differences in the hypothalamic response to stress. The stress-dependent ISIAH strain-specific induction of *Fos* and *Jun* gene transcription may play a crucial role in neuronal activation in this rat strain. The data obtained can be potentially useful in the selection of molecular targets for the development of pharmacological approaches to the correction of stress-induced pathologies related to neuronal excitability, taking into account the hypertensive status of the patients.

## 1. Introduction

Hypertension is a multifactorial disease. Many studies report the existence of an association between psychosocial stress and hypertension [1,2,3]. However, the study of the genetic basis of the increased blood pressure (BP) response to psychoemotional stress remains an urgent problem to date.

The activation of the hypothalamic–pituitary–adrenal (HPA) axis in response to stress stimulates the synthesis of glucocorticoids by the adrenal glands, which affects many processes in the body, including the regulation of BP [4]. The hypothalamus is one of the key brain structures that controls the regulation of glucocorticoid secretion through negative feedback and provides the integration of central and peripheral links involved in BP regulation and the development of hypertension [5,6,7]. The processes associated with hypothalamic activation and systolic blood pressure increase are mediated by sympathetic nerve activation [8].

Researchers often choose the hypothalamus to analyze transcriptomic changes after exposure to various stressors: heat stress for 24 h [9] or for 14 days [10,11], social conflict (within 20 days) [12], alcohol consumption [13], short-term starvation [14], and repeated homotypic stress [15]. The hypothalamic transcriptome has also been investigated when studying the molecular mechanisms of blood pressure regulation. For example, changes in the hypothalamic transcriptome of normotensive Wistar rats were analyzed under hypotonic stress induced by furosemide injection [16].

Transcriptomic changes occurring in the HPA system have also been studied in hypertensive animals. In spontaneously hypertensive (SHR) rats, the effects of manual acupuncture on BP levels and gene transcription in the hypothalamus were studied [17]. A comparative study of the expression profiles of both mRNA and long non-coding RNAs was conducted in the hippocampus of an attention deficit hyperactivity disorder model in spontaneously hypertensive rats and control Wistar Kyoto rats [18]. The effects of a high-salt diet on transcriptome dynamics and on BP were demonstrated in a study of rat hypothalamic and brainstem cardiovascular control centers gene expression profiles [19]. These studies demonstrated that HPA axis responses to stress may depend on the type and duration of stress as well as genotype. In addition, the response to stress may differ between hypertensive and normotensive rats [20].

ISIAH rats with inherited stress-induced arterial hypertension were selected for a sharp increase in systolic BP under short-term (30 min) restraint stress [21,22]. ISIAH rats are a model of a stress-sensitive form of hypertension with genetically pre-determined activation of the HPA and sympathoadrenal systems [23]. The comparative analysis of hypothalamic transcriptomes of hypertensive ISIAH and control normotensive WAG rats allowed us to identify genes related to the inter-strain differences, many of which are associated with both hypertension and peculiarities of central nervous system functioning [24].

Our earlier studies have shown that the temporal dynamics of the increase in plasma corticosterone concentration upon exposure to short-term restraint stress differs in hypertensive ISIAH and normotensive control rats [23]. The analysis of the expression of several key genes associated with the function of the HPA system and kidneys showed that, under conditions of short-term restraint stress, many genes differentially change the level of transcription in the organs/tissues of hypertensive ISIAH compared to normotensive WAG rats [23,25,26,27,28]. In the present study, we sequenced the hypothalamic transcriptomes of hypertensive ISIAH and normotensive WAG rats at rest and after exposure to a single short-term (2 h) restraint stress to reveal the genome-wide patterns of strain-specific neural networks providing differences in hypothalamic response to stress in hypertensive ISIAH and normotensive WAG rats.

## 2. Results

### 2.1. Blood Pressure Level and Plasma Corticosterone Concentration

The measurement of basal blood pressure and blood pressure levels after exposure to short-term (2 h) restraint stress showed significant inter-strain differences in resting BP levels. A significant increase in BP levels upon exposure to stress was shown in ISIAH rats but not in WAG rats (Figure 1a). The plasma corticosterone concentration was significantly increased in both rat strains, indicating the presence of a stress response in both ISIAH and WAG rats (Figure 1b).

### 2.2. Transcriptomic Changes in the Hypothalamus of Hypertensive ISIAH and Normotensive WAG Rats

The altered transcription of many genes is observed in the hypothalamus of both rat strains when exposed to a single restraint stress for 2 h. A total of 257 differentially expressed genes (DEGs) were found in hypertensive ISIAH rats, and 229 DEGs were found in WAG rats. Among them, 144 DEGs are common to rats of both strains (Figure 2a). Accordingly, it can be concluded that there is both a general response to stress and a response specific to each rat strain. The observed changes in the transcriptome of ISIAH rat strain are characterized by the reduction in the transcription levels of most of genes (Figure 2b). The common response includes 144 DEGs, of which 42 genes changed the transcription level more than 1.5-fold in the hypothalamus of both rat strains, and the remaining 102 DEGs changed the transcription level more than 1.5-fold in the hypothalamus of only one of the strain (Figure 2b).

### 2.3. ISIAH Specific 113 DEGs (Appendix A)

A total of 12 DEGs encoding transcription factors (TFs) (Table 1) and 15 DEGs associated with hypertension (Table 2) were identified in the list of 113 DEGs representing a specific stress response in the hypothalamus of ISIAH rats. Two of them (*Ar* and *Fos*) encode TFs associated with hypertension. Most of the 113 genes (103 DEGs, 91.1%) had decreased transcription levels under stress, and only 10 DEGs were activated (Table 3). Among the upregulated DEGs, three genes encoding TFs and three genes associated with hypertension (according to the Rat Genome Database) were found.

### 2.4. Functional Analysis of DEGs Representing Strain-Specific Response to Stress in ISIAH Rats

The functional analysis of 113 DEGs representing strain-specific response to stress in ISIAH rats revealed numerous biological processes involved in the response to short-term restraint stress. The main ones are shown in Figure 3. Notably, in all of the biological processes identified, a large proportion of genes have reduced transcription levels upon exposure to stress. The functional analysis demonstrates that the most statistically enriched terms are associated with transmembrane ion transport, primarily potassium, but also sodium and chlorine (Figure 3). Many terms are associated with the response to various stimuli. In addition, changes relate to cell adhesion and signaling, developmental processes, regulation of metabolic processes, and homeostasis. Below, we review in more detail the terms response to stimulus, ion transport, and signaling, which, from our point of view, most effectively indicate key links in the strain-specific response to stress in the hypothalamus of hypertensive ISIAH rats.

#### 2.4.1. Ion Transport

A total of 25 DEGs were classified into the ion transport group, 4 of which are associated with hypertension (Figure 4). In this group of DEGs, only three genes were activated under stress (*Slc4a1*, *Kel*, and *Apoa1*). The terms associated with molecular functions reflect the activity of various ion channels, such as monocarboxylate:sodium symporter activity, bicarbonate transmembrane transporter activity, potassium channel activity, solute:sodium symporter activity, chloride transmembrane transporter activity, and sodium ion transmembrane transporter activity. It is important to emphasize that the functioning of many of them is related to the presynaptic area.

Terms characterizing biological processes associated with ion transport show that, in the hypothalamus of ISIAH rats, the most significant changes are associated with potassium ion transmembrane transport and with the regulation of ion transmembrane transport (Figure 4). Among the 10 genes associated with potassium ion transmembrane transport, there are DEGs responsible for both the potassium ion export across plasma membrane and potassium ion import across plasma membrane (Appendix A). Three DEGs are involved in the process of regulation of potassium ion transmembrane transport, one of which (*Kel*) is activated under stress.

Accordingly, it can be concluded that, upon exposure to restraint stress in the hypothalamus of ISIAH rats, significant changes in ion transmembrane transport are observed, which most strongly affect the processes associated with potassium ion transmembrane transport. However, the functional analysis also revealed statistically highly significantly enriched terms associated with sodium ion transport (sodium ion transmembrane transport) and chloride transmembrane transport (Figure 3 and Figure 4).

#### 2.4.2. Response to Stimulus

The functional analysis showed that the most numerous group of DEGs is associated with the term response to stimulus (Figure 3). Only 7 of 66 DEGs in this group are activated upon exposure to stress in the hypothalamus of ISIAH rats. Two of them (*Fos* and *Egr4*) encode transcription factors. Among the upregulated DEGs associated with response to stimulus (Table 4), three genes are associated with arterial hypertension. The group of genes associated with response to stimulus includes several subgroups of DEGs. Two of them, response to calcium ion and response to endogenous stimulus, are discussed in more detail below.

##### Response to Calcium Ion

It is known that, when an electrical impulse is delivered to the presynaptic membrane, Ca^2+^ ions enter the presynapse through specialized calcium channels, which triggers the release of mediators into the synaptic cleft and activation of synaptic transmission. The functional analysis of 113 DEGs (Figure 3) allowed us to identify a group of 5 DEGs associated with the response to calcium ion (Table 5), among which two genes, *Fos*, encoding a TF, and *Alox15*, which encodes a member of the lipoxygenase family of proteins, are activated in the hypothalamus of ISIAH rats under the influence of restraint stress.

##### Response to Endogenous Stimulus

Response to endogenous stimulus is represented by several groups of genes associated with response to peptide and steroid hormone stimuli: response to angiotensin, response to gonadotropin, response to progesterone, response to corticosteroid, and response to mineralocorticoid (Figure 5). Only 2 of the 22 genes shown in Figure 5 (*Fos* and *Prlh*) are activated by restraint stress in the hypothalamus of ISIAH rats, while the remaining genes have reduced transcription levels. Seven of DEGs are associated with hypertension.

According to the reviewed results of the functional annotation of DEGs associated with strain-specific response to stimuli in the hypothalamus of hypertensive ISIAH rats, it can be concluded that the activation of transcription factors *Fos* and *Egr4* may play a key role in the regulation of this process.

#### 2.4.3. Signaling

The detailed analysis of DEGs associated with signaling suggests the presence of both negative and positive regulation of signaling. In addition, the analysis revealed the significance of the regulation of MAPK (mitogen-activated protein kinase) cascade, enzyme-linked receptor protein signaling pathway, cell surface receptor signaling pathway, and regulation of protein phosphorylation (Figure 6). According to the involvement of DEGs in the signaling pathways (Figure 6), it can be suggested that the *Ar* (androgen receptor) gene encoding a transcription factor and the *Grm5* (glutamate metabotropic receptor 5) gene may play a key role in these processes. Both of these genes reduce the level of transcription in the hypothalamus of ISIAH rats under restraint stress. Ten DEGs assigned to the signaling group are associated with hypertension.

### 2.5. WAG Specific 85 DEGs (Appendix A)

Among 85 DEGs representing a specific response to restraint stress in the hypothalamus of normotensive WAG rats, 49 genes (57.6%) increase transcription levels upon exposure to stress. We identified seven genes encoding transcription factors (Table 6) and nine genes associated with arterial hypertension (Table 7). Two DEGs (*Esr1* and *Hif3a*) encoding transcription factors and associated with hypertension were identified.

### 2.6. Functional Analysis of DEGs Representing Strain-Specific Response to Stress in WAG Rats

The functional analysis of 85 DEGs revealed numerous biological processes involved in the response to exposure to short-term restraint stress. The main ones are shown in Figure 7. The functional analysis demonstrated that the most statistically enriched terms were associated with protein phosphorylation, signaling, ion transport, and response to various stimuli. Also, the changes concerned the regulation of metabolic processes and the maintenance of homeostasis.

The results of the functional analysis for the groups of genes associated with the terms ion transport, response to endogenous stimulus, and signaling are shown in more detail (Figure 8, Figure 9 and Figure 10).

#### 2.6.1. Ion Transport

In the hypothalamus of WAG rats, 20 genes are assigned to the group of DEGs associated with term ion transport. The transcription of 11 of them is activated, and 9 genes decrease the level of transcription under stress. Four DEGs are associated with hypertension. The functional analysis of these DEGs allow the specification of changes in ion channel activity. (Further in the text, downregulated genes are printed in blue and upregulated genes are printed in red) Groups of genes controlling acetylcholine-gated cation-selective channel activity (*Chrna9*, *Chrna10*), chloride channel activity (*Best3*, *Gabrd*, *Gabrq*), calcium channel activity (*Chrna9*, *Chrna10*, *Trpm5*), and potassium channel activity (*Kcnj2*, *Kcnk15*, *Trpm5*) were identified. The functional analysis of DEGs associated with ion transport emphasized that nine of them exert postsynaptic regulation of ion transport (Figure 8).
Figure 8Functional annotation of 20 WAG strain-specific DEGs associated with ion transport. Purple lines indicate experimentally determined interactions; blue lines denote known interactions from curated databases; dark blue lines represent gene co-occurrence; black lines indicate co-expression; and green lines represent the results of text mining. Protein–protein interaction (PPI) enrichment *p*-value: 0.0248. *, genes associated with hypertension.
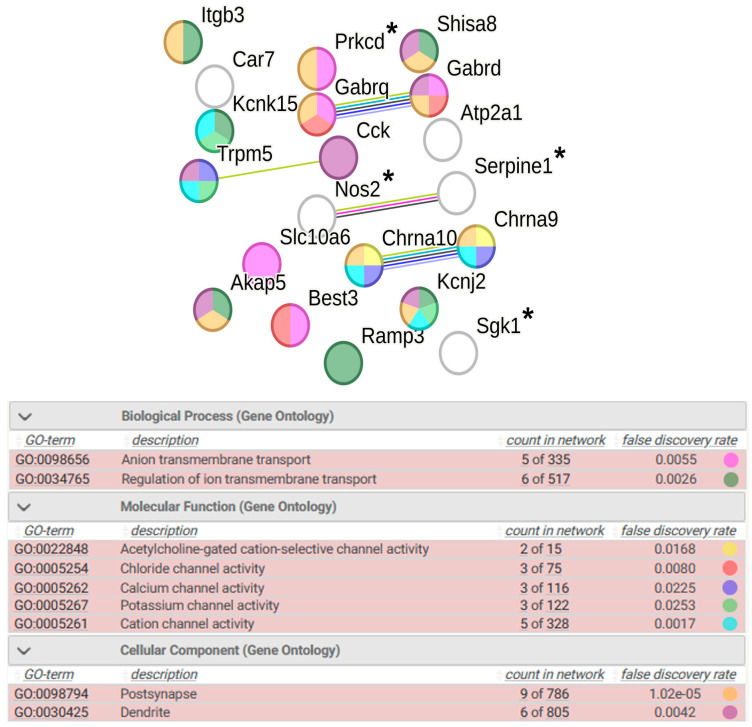


#### 2.6.2. Response to Endogenous Stimulus

The functional analysis of DEGs associated with the term response to endogenous stimulus allowed us to understand which stimuli have a significant effect on the level of gene transcription under conditions of restraint stress in the hypothalamus of WAG rats. The groups of DEGs associated with cellular response to aldosterone, to platelet-derived growth factor stimulus, to vitamin D, to angiotensin, to growth hormone, to glucose, to insulin stimulus, to glucocorticoid, and to estradiol were identified (Figure 9). According to the relationships shown in Figure 9, the greatest number of effects are mediated through changes in the transcription level of the genes *Sgk1*, *Esr1*, *Nos2*, *Serpine1*, *Gh1*, and *Errfi1*. The first four of these genes are associated with hypertension, with the *Esr1* gene encoding a TF.
Figure 9Functional annotation of 22 WAG strain-specific DEGs associated with response to endogenous stimulus. Purple lines indicate experimentally determined interactions; blue lines denote known interactions from curated databases; dark blue lines represent gene co-occurrence; black lines indicate co-expression; and green lines represent the results of text mining. Protein–protein interaction (PPI) enrichment *p*-value: 1.64 × 10^−8^. *, genes associated with hypertension.
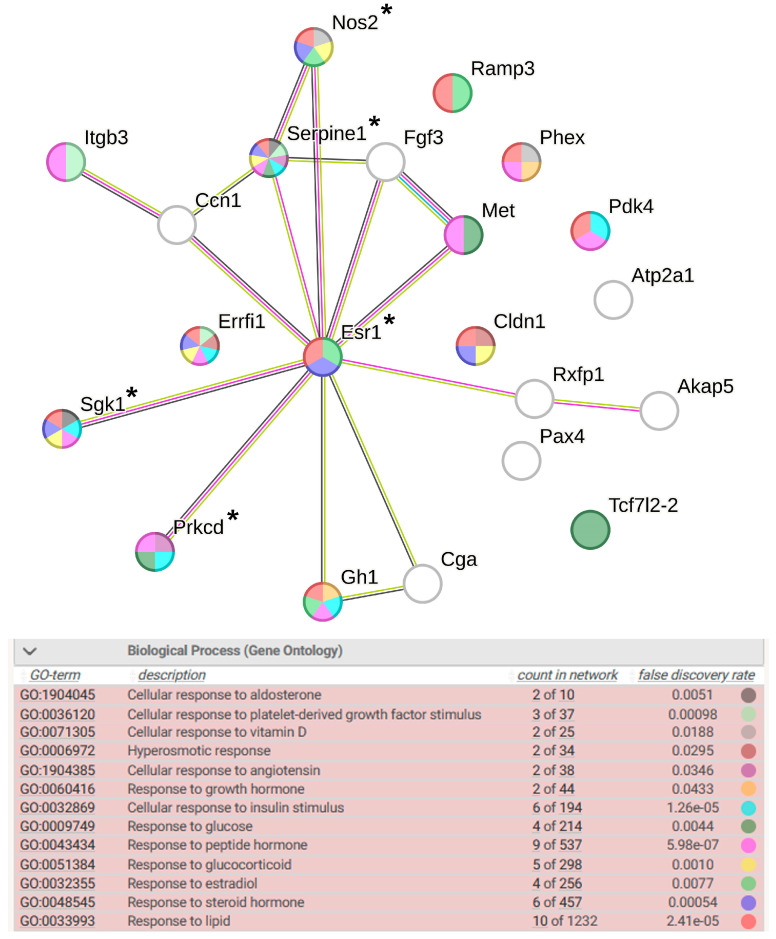


#### 2.6.3. Signaling

The detailed analysis of DEGs associated with signaling revealed the importance of regulation of ERK1 and ERK2 cascade, regulation of MAPK cascade, transmembrane receptor protein tyrosine kinase signaling pathway, cell surface receptor signaling pathway, positive regulation of signaling, G protein-coupled receptor signaling pathway, regulation of protein phosphorylation, and regulation of signaling (Figure 10). According to the involvement of DEGs in signaling pathways, it can be assumed that the key role in these processes may be played by the genes *Adra2b, Ccl9*, *Ccl24*, *Ccn1*, *Esr1*, *Fzd5*, *Gh1*, *Itgb3*, *Met*, and *Ramp3*, three of which are associated with hypertension, and the *Esr1* gene encodes a transcription factor.
Figure 10Functional annotation of 49 WAG strain-specific DEGs associated with signaling. Purple lines indicate experimentally determined interactions; blue lines denote known interactions from curated databases; dark blue lines represent gene co-occurrence; black lines indicate co-expression; and green lines represent the results of text mining. Protein–protein interaction (PPI) enrichment *p*-value: 1.1 × 10^−13^. *, genes associated with hypertension.
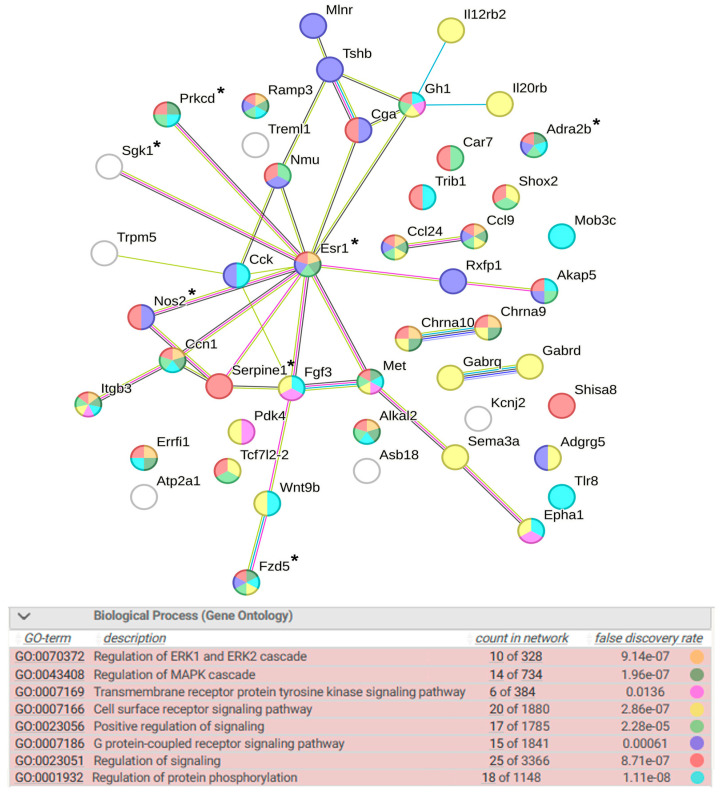


### 2.7. The Common Response to Stress for the Two Rat Strains (Appendix A)

The common response to stress for the two rat strains is represented by a group of 144 DEGs. Among them, 42 genes changed the level of transcription equally strongly (more than 1.5-fold) in the hypothalamus of both rat strains. One of these 42 genes (*Npas4*) changed the transcription level differently in the hypothalamus of ISIAH and WAG rats when exposed to stress (Figure 11). The remaining 102 genes changed statistically significantly and unidirectionally in both rat strains, but the changes in their transcription levels exceeded a 1.5-fold threshold in only one of the strains (Appendix A). These 144 DEGs, which most significantly changed the level of transcription in the hypothalamus of both rat strains, are considered as a group of genes of the general response to stress, and functional annotation for them was carried out jointly.

A group of 144 DEGs contains 18 genes encoding transcription factors (Table 8). Seven of them are activated by stress. A group of 144 DEGs identified 20 genes associated with hypertension (Table 9). Most of the DEGs associated with hypertension reduce transcription levels when exposed to stress (only three of them are activated). Two DEGs (*Irf8* and *Smad9*) are associated with hypertension and encode TFs (both are downregulated).

### 2.8. Functional Analysis of 144 Common DEGs

The functional analysis of 144 DEGs revealed common biological processes involved in the hypothalamic response to exposure to a single short-term restraint stress in rats of both strains. The main ones are shown in Figure 12.

The functional analysis demonstrates that the most statistically enriched terms are associated with the regulation of developmental process, signaling, response to stimulus, cell adhesion, and cell differentiation. Also, the changes concern biologically relevant processes, such as adenosine triphosphate (ATP) generation from ADP, reactive oxygen species metabolic process, inflammatory response and its regulation, and transport. In Figure 12, the asterisk indicates the biological processes with which the *Npas4* gene, which alternatively changes the level of transcription in the hypothalamus of hypertensive ISIAH and normotensive WAG rats, is associated. According to the functional analysis, we can conclude that *Npas4* can alternatively affect signaling and response to stimulus processes, including those related to the response to corticosterone. We will discuss in more detail below the groups of DEGs associated with signaling, response to stimulus, and transport, in particular ion transport.

#### 2.8.1. Signaling

More than half of the common DEGs (77 DEGs) are involved in processes associated with signaling. Twelve of these DEGs are associated with hypertension (Figure 13). According to the functional annotation of these DEGs, the most significant roles are the regulation of adenylate cyclase-activating adrenergic receptor signaling pathway involved in heart process, the positive regulation of endothelial cell chemotaxis by VEGF-activated vascular endothelial growth factor receptor signaling pathway, toll-like receptor 2 signaling pathway, the positive regulation of integrin-mediated signaling pathway, vascular endothelial growth factor signaling pathway, positive regulation of pattern recognition receptor signaling pathway, regulation of G protein-coupled receptor signaling pathway, the regulation of transforming growth factor beta receptor signaling pathway, the positive regulation of transmembrane receptor protein serine/threonine kinase signaling pathway, and MAPK cascade (Figure 13). It can be assumed that the regulation of phosphorylation plays a significant role in the regulation of signaling. NPAS4 is involved in the regulation of cell surface receptor signaling pathway; accordingly, the regulation of these processes may be achieved differently in the hypothalamus of ISIAH and WAG rats.

#### 2.8.2. Response to Endogenous Stimulus

There are 35 DEGs involved in response to endogenous stimulus. Their transcription level can change in response to physical factors (ischemia, hypoxia, and temperature stimulus), in response to fatty acids, lipids, cytokines, growth factor, amyloid-beta, in response to epinephrine, as well as in response to steroid hormones (corticosterone and progesterone) (Figure 14). NPAS4 is involved in response to corticosterone and, therefore, may have different effects on the formation of response to endogenous stimulus in the hypothalamus of ISIAH and WAG rats. Nine genes are associated with hypertension in this group of DEGs.

#### 2.8.3. Transport

The transcription of 53 DEGs associated with transport processes is altered in the hypothalamus of both rat strains (Figure 12). The functional annotation of these DEGs identified a group of 16 DEGs associated with ion transport (FDR 1.61 × 10^−5^) (Figure 15). Of these, 12 DEGs were associated with ion transmembrane transporter activity, which may reflect changes in icosanoid transport, calcium ion transport, and carboxylic acid transport. Among the 16 DEGs associated with ion transport, the expression of only one gene (*Pla2g3*) is activated by stress. Changes in ion transmembrane transporter activity occur in both pre- and postsynaptic membranes. These processes common to the two rat strains affect glutamatergic synapse function. Genes associated with cellular response to acetylcholine may also play a significant role in ion transport processes. Two of them (*Chrna7* and *Chrm5*) reduce the level of transcription, altering acetylcholine receptor activity. Four DEGs responsible for ion transport are associated with hypertension.

### 2.9. The Alternative Role of NPAS4

In the resting state, *Npas4* gene expression in the hypothalamus of ISIAH rats was statistically significantly decreased compared to the control WAG rats (Figure 16). When exposed to restraint stress for 2 h, there is activation of *Npas4* expression in the hypothalamus of hypertensive ISIAH rats, and the level of *Npas4* gene transcription in the hypothalamus of WAG rats decreases and becomes equal to the level of *Npas4* transcription in the hypothalamus of ISIAH rats at rest.

Querying NPAS4 using the Enrichr-KG resource (Figure 17) indicates that alternative changes in the transcription level of the *Npas4* gene in the hypothalamus of ISIAH and WAG rats may be a key event in the regulation of GABAergic synaptic transmission, both through the inhibition and excitation of the postsynaptic potential. According to the analysis presented in Figure 17, NPAS4 may be a key link associated with increased susceptibility to neuronal excitotoxicity and neuron degeneration.

The correlation analysis of *Npas4* gene expression was performed using all DEGs. The first 20 most significant correlations of *Npas4* gene expression with other DEGs are summarized in Table 10. It can be seen that *Npas4* gene expression correlates with different sets of genes in the hypothalamus of ISIAH and WAG rats. In the hypothalamus of ISIAH rats, *Npas4* gene expression correlates statistically most significantly with the expression of *Jun* and *Fos* genes, which are associated with hypertension and encode TFs. In the hypothalamus of WAG rats, *Npas4* gene expression correlates statistically most significantly with the expression of other DEGs, among which the *Bcl2l1* gene is associated with hypertension and four DEGs encode TFs (*Isl1*, *Hnrnpr*, *Mlxipl*, and *Zfp189*).

Querying the DEGs presented in Table 10 indicates the most functionally relevant genes that may be associated with the hypothalamic response to exposure to restraint stress in ISIAH rats (Figure 18) and in WAG rats (Figure 19), as well as the biological processes and metabolic pathways with which these DEGs are associated. The comparison of changes in the transcription level of these DEGs in the hypothalamus of ISIAH and WAG rats (Appendix A) shows that the most significant inter-strain differences in stress response are associated with the *Fos* and *Jun* genes The expression of *Fos* and *Jun* genes is induced only in the hypothalamus of hypertensive ISIAH rats, but not WAG (Figure 16b,c).

According to the results obtained in our study, we can conclude that alternative changes in the level of expression of the *Npas4* gene in the hypothalamus of ISIAH and WAG rats may be a key link in the strain-dependent response to stress. The ISIAH strain-specific response to stress is associated with the activation of transcription of the *Fos* and *Jun* genes encoding TFs.

## 3. Discussion

In the presented study, a comparative analysis of transcriptome response to a single short-term (2 h) restraint stress in the hypothalamus of hypertensive ISIAH and normotensive WAG rats was performed. A large number of DEGs were found to alter transcription levels upon exposure to stress in the hypothalamus of both rat strains. A group of 144 DEGs characterizing a common stress response for the two rat strains was identified, as well as groups of DEGs describing a strain-specific hypothalamic response to stress in hypertensive ISIAH rats (113 DEGs) and normotensive WAG rats (85 DEGs). Thus, comparison of transcriptional profiles shows significant inter-strain differences in the response to restraint stress, both in the lists of DEGs and in the functional groups represented by these genes.

Many genes associated with hypertension are involved in the stress response. They are present among DEGs related to both common and strain-specific responses to stress. Accordingly, despite the fact that a significant increase in blood pressure under stress is observed only in ISIAH rats, it can be assumed that exposure to stress triggers blood pressure regulation mechanisms in the hypothalamus of rat of both strains. Strain-specific differences in stress response were expected and are likely due to genetically determined differences in stress sensitivity of hypertensive ISIAH and normotensive WAG rats [23]. The identified strain-specific features of BP regulation under stress in hypertensive ISIAH and normotensive WAG rats create the basis for further research aimed at finding targets for pharmacological intervention in order to effectively regulate BP levels under the influence of emotional stress, depending on the hypertensive status of patients.

The functional annotation of DEGs associated with the overall stress response showed that the hypothalamus of both rat strains exhibited a response to various endogenous stimuli of different nature: lipids, cytokines, growth factor, amyloid-beta, epinephrine, and steroid hormones. This result demonstrated and once again confirmed the classical ideas about the participation in the stress response of the listed factors [29,30] and steroid hormones (corticosterone and progesterone) [31,32], the influence of which is manifested independently of the type of stressor exposure. In our experiment, the response to these factors was independent of the genotype of the compared rat strains. Another group of genes, which from our point of view deserves special attention, is the group of DEGs associated with the response to hypoxia. All genes in this group change the transcription level unidirectionally in hypertensive ISIAH and normotensive WAG rats, confirming the development of a brain hypoxia state independently of the hypertensive genotype. Cerebral hypoxia–ischemia can cause a wide range of biological responses, such as cell membrane depolarization, excitotoxicity, oxidative stress, inflammation, and apoptosis [33]. The results of the functional annotation of common DEGs performed in our study (Figure 12) are in good agreement with these views. Also, our results are in good agreement with studies of various hypoxia-related pathologies. It is believed that hypoxia-induced signaling pathways may regulate adaptive responses, such as endoplasmic reticulum (ER) stress response, anti-oxidative responses, and autophagy, and represent the underlying mechanisms of pathogenesis of various diseases [34,35]. Accordingly, our findings, which describe a common response to the restraint stress used in our study, may be useful for understanding the general molecular genetic mechanisms associated with changes in gene transcription levels in response to different types of stress.

The main objective of our study was to identify specific mechanisms of stress response in hypertensive ISIAH rats characterized by increased sensitivity to stress. When considering the general stress response, we identified a gene *Npas4* (neuronal PAS domain protein 4) encoding the TF, which changed the level of transcription differently in the hypothalamus of ISIAH and WAG rats.

Earlier, when studying the effect of *Npas4* expression on excitatory synaptic activity, it was shown that abolishing *Npas4* expression in a neuron increases the number and/or presynaptic release probability of excitatory synapses that form on the neuron. On the contrary, *Npas4* activation is associated with an increase in synaptic inhibition and a decrease in the excitation of a neuron, which can be considered as a negative feedback mechanism to maintain the homeostatic balance between neuronal excitation and inhibition [36]. Accordingly, Npas4 is considered as an early-response TF that regulates the expression of inhibitory synapse genes to control homeostatic excitatory/inhibitory balance in neurons [36,37]. According to this information, we can hypothesize that, in the hypothalamus of WAG rats, the decreased transcription levels of the *Npas4* gene under stress may indicate the activation of synaptic transmission. The activation of *Npas4* in the hypothalamus of ISIAH rats may act to diminish the excitatory synaptic input.

The functional annotation of DEGs in our study showed that *Npas4* is involved in various biological processes (Figure 12), including response to corticosterone. It has been previously reported that the expression of *Npas4* in the mouse hippocampus decreased under the conditions of chronic or acute (for 3 h) restraint stress, as well as during the acute administration of corticosterone [38,39]. It has been shown that the reduction in *Npas4* expression occurs through the binding of agonist-bound glucocorticoid receptors (GRs) to glucocorticoid response elements (GREs) sequences localized in the promoter region of *Npas4* [39]. Given that this mechanism of regulation of *Npas4* expression has been shown in the above studies in normotensive animals, we hypothesize that the decrease in *Npas4* transcription levels under stress and the possible activation of synaptic transmission in the hypothalamus of WAG rats may follow this scenario. Given that ISIAH rats are characterized by the genetically pre-determined activation of the hypothalamic-pituitary–adrenal and sympathoadrenal systems [23], we can assume that this mechanism provides an initially reduced basal level of *Npas4* expression in the hypothalamus of hypertensive ISIAH rats. Thus, our results allow us to conclude that the *Npas4* gene is one of the key gene regulators that may be responsible for the state of genetically determined increased activation of the sympathetic nervous system in hypertensive ISIAH rats.

The alternative change in the level of *Npas4* transcription in the hypothalamus of hypertensive ISIAH rats compared to normotensive WAG rats suggests the involvement of mechanisms related to the peculiarities of the physiological state of ISIAH rats in the stress response. *Npas4* expression is known to be induced in response to the stimulation of excitatory synaptic activity by a glutamate or by a general non-selective central nerve system stimulating agent Metrazol, known to induce immediate early-type genes in brain [40]. The activation of *Npas4* was reported in ischemic rat brain following middle cerebral artery occlusion. The observed increase in *Npas4* expression was directly correlated with the severity of neuronal damage [41]. It has been shown that perturbations in systems using the excitatory amino acid L-glutamate may cause the glutamate-mediated excitotoxicity and underlie the pathogenic mechanisms of hypoxia–ischemia and a plethora of neurological disorders [42]. According to the results of several studies, Npas4 induction is considered as a neuroprotective factor and, accordingly, *Npas4* has been characterized as a gene responsible for neuroprotection induced by synaptic activity [43,44,45,46]. At the same time, *Npas4* induction can be observed not only in response to hypoxia–ischemia but also to other types of stress—experience-dependent activation [47], exposure to foot-shock apparatus (moderate stressor intensity), and application of foot-shocks (high stressor intensity) [48]. In the light of these ideas, the alternative change in *Npas4* transcription in our study can be interpreted as a factor balancing the level of brain neuronal activation near a set-point that provides the necessary level of brain activity to maintain a characteristic value of neuronal activity for each rat strain.

The induction of *Npas4* has been characterized as an immediate early-type gene expression mechanism [40]. Our data are in good agreement with these notions. In the hypothalamus of ISIAH rats, *Npas4* gene expression correlates statistically most significantly with the expression of the *Fos* and *Jun* genes, which encode subunits of the AP-1 transcription regulator. They belong to immediate early genes (IEGs), as they are rapidly induced in the brain by a variety of stimuli. Fos activation is a recognized marker of neuronal activation [49,50]. We have previously shown that, in the hypothalamus of ISIAH rats, the dynamics of *Fos* gene expression after a single exposure to restraint stress of different durations (30, 60, and 120 min) coincides with the dynamics of the increase in blood pressure in these rats [51]. As seen in Figure 16, the expression of *Fos* and *Jun* genes is induced only in the hypothalamus of hypertensive ISIAH rats, but not WAG, which coincides well with the change in blood pressure levels in rats of these strains (Figure 1a). Accordingly, we can conclude that the sharp increase in the blood pressure level characteristic of ISIAH rats upon exposure to brief restraint stress may be associated with the activation of hypothalamic neurons, which is mediated by the activation of *Fos* and *Jun* gene expression. Since *Fos* activation indicative of neuronal activation upon exposure to a 2-h restraint stress is observed only in the hypothalamus of hypertensive ISIAH rats, but not WAG, it can be assumed that the type of stress used in our study is not a strong enough to activate the hypothalamic neurons of control WAG rats. On the other hand, given that ISIAH rats are characterized by genetically pre-determined increased stress reactivity [23], it can be assumed that different time periods are required for the development of the stress response in ISIAH and WAG rats. Accordingly, the inter-strain differences in the stress response may be related to both the genetically pre-determined hypertensive state of ISIAH rats and their increased stress reactivity compared to the control rats [23]. Anyway, the alternative changes of *Npas4* transcription levels in the hypothalamus of ISIAH and WAG rats revealed in our work require additional studies, since different pathways of the activation of this TF imply not only a diversity of results of this activation [47,52], but also alternative mechanisms for implementing its regulatory programs [53,54].

A total of 113 DEGs representing a specific response to restraint stress were identified in the hypothalamus of ISIAH rats in our experiment. Importantly, the vast majority of these DEGs had decreased transcription levels upon exposure to stress. The functional analysis demonstrates that the most statistically enriched terms are associated with transmembrane ion transport processes, primarily potassium transmembrane ion transport (Figure 3, Figure 4 and Appendix A).

Voltage-gated potassium channels represent the complex class of voltage-gated ion channels. One of their various functions is the regulation of neuronal excitability [55]. As can be seen from Figure 3, in the hypothalamus of ISIAH rats, the levels of transcription of a significant number of genes encoding voltage-gated K+ channel subunits reduced. It was shown that gain-of-function mutations in genes encoding voltage-gated potassium channels can increase the excitability of neurons and vice versa, loss-of-function mutations reduce action potential firing frequency (reviewed in [56]). Accordingly, we can suggest that the downregulation of the expression of several genes encoding the subunits of voltage-gated potassium channels is aimed at reducing neuronal excitability, which probably exceeded the physiologically adequate threshold of neuronal excitation in the hypothalamus of ISIAH rats under restraint stress and, like *Npas4* activation, contributes to neuroprotection.

In the hypothalamus of WAG rats, the strain-specific downregulation of two genes encoding voltage-gated potassium channels was also found (Figure 8). Accordingly, it can be assumed that the 2-h restraint stress used in our study induces neuronal excitability not only in the hypothalamus of ISIAH rats but also in the hypothalamus of the control WAG rats, but the degree of neuronal excitability changes in the hypothalamus of WAG rats is much weaker.

Within the scope of this manuscript, we discussed only the most relevant, from our point of view, molecular genetic events reflecting general and specific responses to restraint stress. It is not possible to discuss in detail all the molecular genetic events reflecting changes in biological processes presented in the results section in a single manuscript. In addition, effects described by less significant changes in gene transcription levels (less than 1.5-fold) were left outside the scope of this manuscript. Nevertheless, we hypothesize that less pronounced changes in gene transcription levels may also have meaningful physiological effects in regulating the response to the proposed stress challenge. The authors anticipate discussing these effects further in separate manuscripts.

The limitations of our study include the fact that the results, while suggesting a significant role of the *Npas4* gene in the regulation of the response to stress, do not allow us to associate changes in its activity with hypertension. The association of *Npas4* gene expression with hypertension has not yet been established in other hypertensive animal models. However, there are studies showing that *Npas4* polymorphisms contribute to the cardiovascular diseases risk (coronary heart disease) [57]. *Npas4* has also been shown to increase transcription levels in the brain under conditions of dehydration [58]. It is known that dehydration has a profound influence on neuroexcitability, but on the other hand, it is also known that water deprivation can change the water–salt balance of the body. An increase in the concentration of sodium ions in body fluids activates the sympathetic neural activity leading to hypertension [59]. Thus, it can be hypothesized that *Npas4* may be involved in the regulation of BP. This aspect may be very promising for a further study of the role of *Npas4* not only as a factor influencing neuroexcitability, but also as an important link in the formation of hypertensive status.

## 4. Materials and Methods

### 4.1. Animals

The experiment was carried out on three-month-old male hypertensive rats of the ISIAH/Icgn (inherited stress-induced arterial hypertension) strain and normotensive rats of the WAG/GSto-Icgn (Wistar Albino Glaxo) strain. Both the hypertensive ISIAH and normotensive WAG rat strains were derived from closely related outbred populations of Wistar Albino rats. We previously characterized WAG rats as a strain with lower behavioral, hormonal, and arterial responses to emotional stress compared to ISIAH rats [21,22], allowing the use of WAG rats as a suitable control group in experiments with hypertensive ISIAH rats. The rats were kept on the basis of the conventional vivarium of the Center for Genetic Resources of Laboratory Animals of the Institute of Cytology and Genetics, Siberian Branch of the Russian Academy of Sciences, where the animals were kept under standard conditions, without restrictions in water and balanced food, and the light regime of the day was 12:12.

For hypothalamic transcriptome analysis by RNA-Seq, 4 groups of 7 animals each were formed: (1) ISIAH_control; (2) WAG_control; (3) ISIAH_stress; and (4) WAG_stress. In all rats, the basal systolic blood pressure (BP) was measured by the tail-cuff method [23] under light ether anesthesia to prevent emotional stress during the measurement. Seven days after basal BP measurement, a 2-h restraint (emotional) stress was performed for rats of two experimental groups, ISIAH_stress and WAG_stress. The stressing procedure consisted of placing the rat in a tight wire-mesh cage for 2 h (but the rat’s tail was outside the cage). Thirty min before the end of the stress procedure, the cage with the rat was placed on a warm (37 °C) platform to prepare the rat for blood pressure measurement. This condition of BP measurement is caused by the need to increase the pulsation of the tail artery and is accepted as a standard approach to obtain correct BP results. There were no signs of additional excitement or other negative behavioral reactions in the rats after a slight warming of the environment.

Immediately after the end of the stress procedure, the rats were measured for BP (without anesthesia) and immediately decapitated; the hypothalamus was isolated and homogenized in 700 μL of ExtractRNA reagent (Evrogen, Moscow, Russia) using 500 μL of Lysing matrix D (Cat#6540434 MP Biomedicals, Solon, OH, USA) for 20 s at 18,000 rpm in a Super FastPrep-2 homogenizer (MP Biomedicals, Solon, OH, USA). All procedures were conducted in compliance with European Communities Council Directive 210/63/EU of 22 September 2010. The study protocol was approved by the Bioethical Council of the federal research center Institute of Cytology and Genetics SB RAS (Novosibirsk, Russia), protocol No. 115 of 20 December 2021.

### 4.2. Measurement of Corticosterone Concentration in Blood Plasma

Plasma corticosterone concentrations were measured using an enzyme-linked immunosorbent assay kit (Ab108821 Corticosterone ELISA Kit, Abcam, Boston, MA, USA), according to the manufacturer’s instructions. Corticosterone concentrations were measured in ng/mL.

### 4.3. RNA-Seq

RNA isolation was performed at the Institute of Genomic Analysis (Moscow, Russia). The sample preparation and sequencing of hypothalamic transcriptomes according to the manufacturer’s protocols (MGI Tech Co., Ltd., Shenzhen, China) were performed at BGI Hongkong Tech Solution NGS Lab (Hong Kong, China). The pair-end sequencing of cDNA libraries was performed on the DNBSEQ platform (DNBSEQ Technology, Hong Kong, China) with a read length of 150 base pairs and sequencing depth of more than 30 million uniquely mapped reads. All samples were analyzed as biological replicates.

The quality of the obtained sequencing data was assessed using the FastQC program (version 0.11.5 [60]). The total number of nucleotide reads for the libraries after filtering was 1,287,393,367, of which 1,267,436,623 nucleotide reads (98.45%) were mapped to the reference rat genome mRatBN7.2/rn7 (rn7 assembly Wellcome Sanger Institute Nov, 2020) using the STAR software package, version 2.7.10a [61].

Statistical analysis to calculate differential gene expression was performed in the R statistical computing environment. We applied surrogate variable analysis (SVA) [62] to account for unwanted variation in the data caused by possible random systematic bias in the sample preparation process. For SVA analysis, expression data were normalized and transformed using the vst function in DESeq2 v1.30.1 [63] according to the documentation. Significant surrogate variables were further included as factors in the differential expression analysis in DESeq2. Differential expression analysis was performed between the comparison groups ISIAH_stress-ISIAH_control (the list of resulting DEGs is designated as ISIAH_DEGs) and WAG_stress-WAG_control (the list of resulting DEGs is designated as WAG_DEGs). Hypothalamic samples from 7 rats were analyzed in each of the 4 groups.

Differential expression was calculated for all genes showing sufficient expression levels above a threshold (sum of gene coverage for all libraries greater than 10 reads). The significance threshold for identifying differentially expressed genes was chosen with correction for multiple comparisons and corresponded to an adjusted *p*-value < 5%. In this manuscript, only DEGs that changed transcription levels under stress by more than 1.5 fold (Log2fold change ≥ |0.585|) were analyzed.

### 4.4. Functional Annotation of DEGs

The functional annotation of DEGs was carried out using the Rat Genome Database [64], DAVID (The Database for Annotation, Visualization and Integrated Discovery) [65], STRING database [66], and the resource Enrichr-KG [67].

### 4.5. Statistical Analysis

Statistical analysis was performed using the Statistica 8.0 software package (StatSoft, Tulsa, OK, USA). A Student’s *t*-test was used to assess intergroup differences in blood pressure levels and corticosterone concentrations in the blood plasma of rats. Data are presented as arithmetic means and their errors (M ± SEM). Pearson’s correlation analysis was performed using two-tailed test.

## 5. Conclusions

The results presented in this manuscript describe the biological processes that are most significantly altered in the hypothalamus of hypertensive ISIAH and normotensive WAG rats in response to a single short-term (2 h) restraint stress. It is shown that the hypothalamic response to stress is represented by some general mechanisms as well as by a strain-specific response that depends on genetically pre-determined features of the physiological state of the compared rat strains. A specific response to stress is associated with a significant number of DEGs in both strains of rats, suggesting differences in the functioning of hypothalamic gene networks in hypertensive and normotensive rats. First of all, it should be noted that the specific response to stress in the hypothalamus of ISIAH rats is associated with a significant decrease in the regulation of most DEGs. Furthermore, it should be noted that the neuronal *Npas4* transcription factor gene was the only one that alternatively significantly changed the expression in the hypothalamus of ISIAH and WAG rats. Strain-specific changes in *Npas4* gene transcription can be considered a key event for understanding inter-strain differences in the hypothalamic response to stress in the analyzed rat strains. In the hypothalamus of WAG rats, *Npas4* expectedly decreased its expression in response to an increase in corticosterone levels. A specific increase in *Npas4* expression during stress-induced increases in corticosterone levels in the hypothalamus of ISIAH rats may reflect the activation of neuroprotection to compensate for excessive neuronal excitation. The identified lineage-specific stress-dependent induction of *Fos* and *Jun* gene transcription indicates neuronal activation in response to stress in hypertensive ISIAH rats, but not in normotensive WAG rats. Considering our earlier data that the induction of *Fos* gene transcription occurs in a coordinated manner with an increase in blood pressure in ISIAH rats, it can be assumed that there is a connection between the features of the functioning of ISIAH strain-specific hypothalamic gene networks described in the present study and a sharp increase in blood pressure in ISIAH rats exposed to stress. The differences in the molecular genetic mechanisms of the response to stress found in the present study may, at least in part, explain the inter-strain differences in changes in blood pressure levels in stressed ISIAH and WAG rats, which are characterized by genetically determined differences in stress sensitivity.

Because the restraint stress we used in our study can be viewed as an emotional stressor, the results presented may be useful in understanding the specific molecular genetic events caused by exposure to short-term emotional stress in normal and hypertension conditions. Potentially, the results obtained in this study may provide new ideas for the development of pharmacological agents to normalize the function of the central nervous system in stress-induced pathologies related to neuronal excitability taking into account the hypertensive status of the patients.

## Figures and Tables

**Figure 1 ijms-25-06680-f001:**
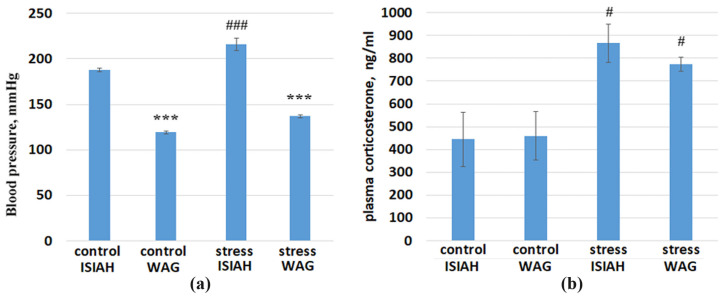
(**a**) Blood pressure levels in ISIAH and WAG rats in the control (measured under light ether anesthesia) and after exposure to stress (measured without anesthesia). (**b**) Plasma corticosterone concentration of ISIAH and WAG rats in the control and after exposure to stress; *n* = 7 in each group. *** *p* < 0.001, inter-strain differences. # *p* < 0.05; ### *p* < 0.001, comparison of the control and after stress exposure.

**Figure 2 ijms-25-06680-f002:**
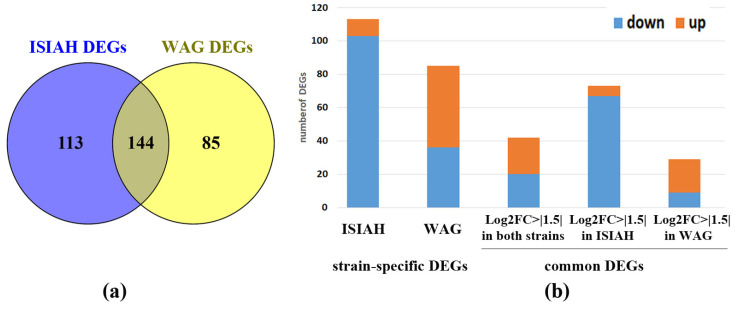
(**a**) Comparison of the number of DEGs. (**b**) Changes in gene transcription levels in the hypothalamus of ISIAH and WAG rats when exposed to single restraint stress for 2 h.

**Figure 3 ijms-25-06680-f003:**
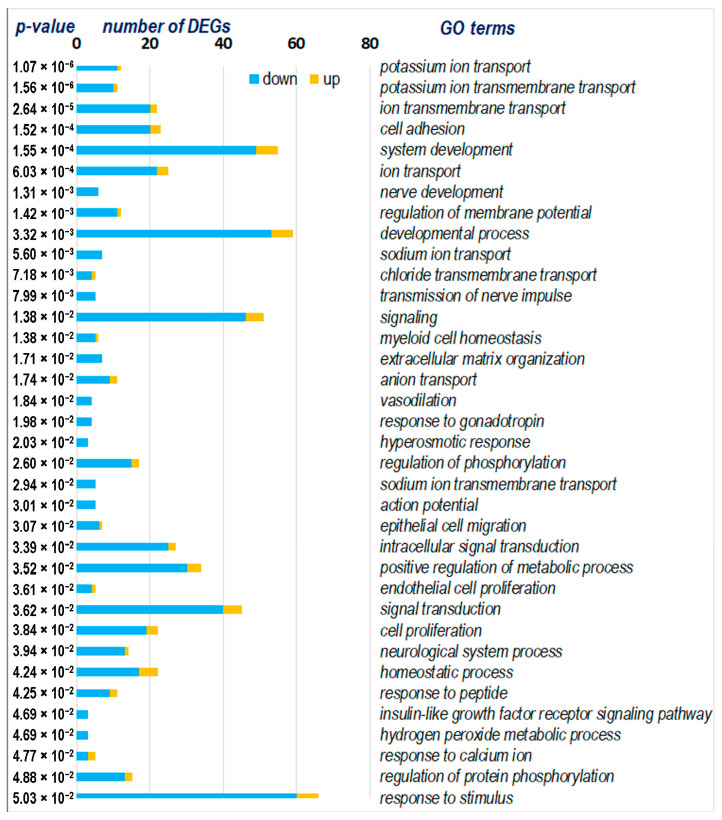
Gene ontology (GO) terms for 113 ISIAH strain-specific DEGs.

**Figure 4 ijms-25-06680-f004:**
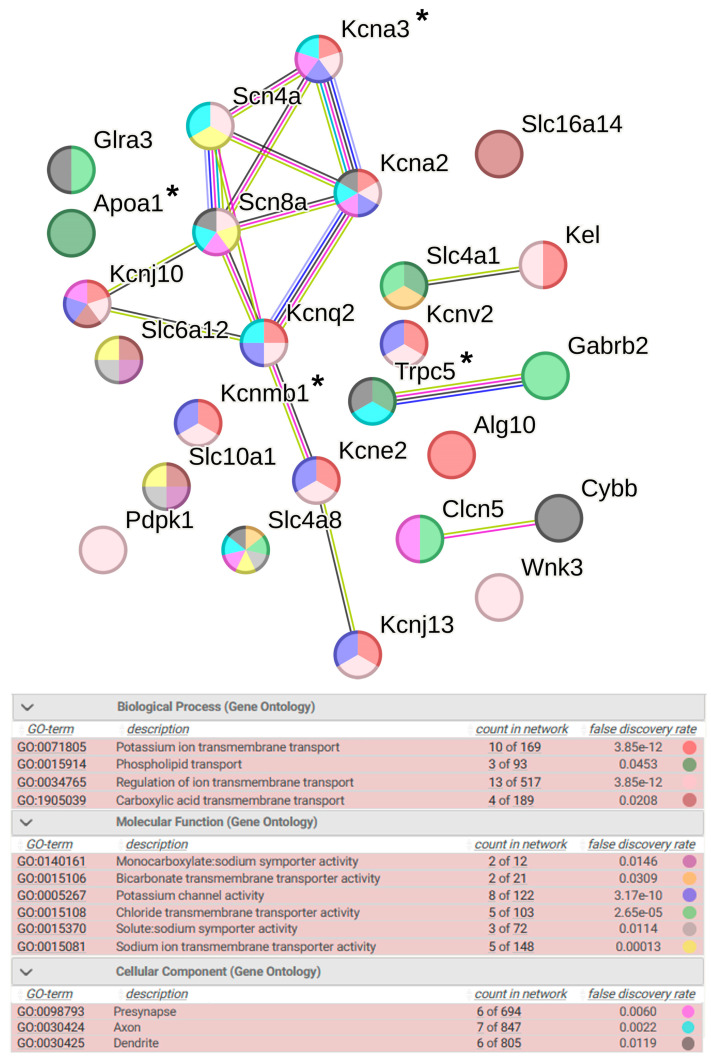
Functional annotation of 25 ISIAH strain-specific DEGs associated with ion transport. Edges represent protein–protein associations. Purple lines indicate experimentally determined interactions; blue lines denote known interactions from curated databases; dark blue lines represent gene co-occurrence; black lines indicate co-expression; and green lines represent results of text mining. Protein–protein interaction (PPI) enrichment *p*-value: 5.11 × 10^−10^. *, genes associated with hypertension.

**Figure 5 ijms-25-06680-f005:**
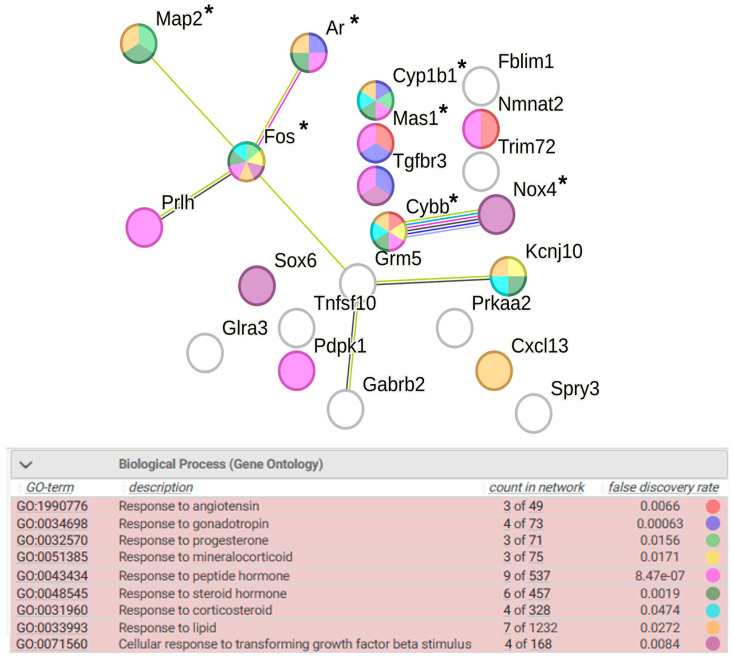
Functional annotation of 22 ISIAH strain specific DEGs associated with response to endogenous stimulus. Edges represent protein–protein associations. Purple lines indicate experimentally determined interactions; blue lines denote known interactions from curated databases; dark blue lines represent gene co-occurrence; black lines indicate co-expression; and green lines represent the results of text mining. Protein–protein interaction (PPI) enrichment *p*-value: 0.00436. *, genes associated with hypertension.

**Figure 6 ijms-25-06680-f006:**
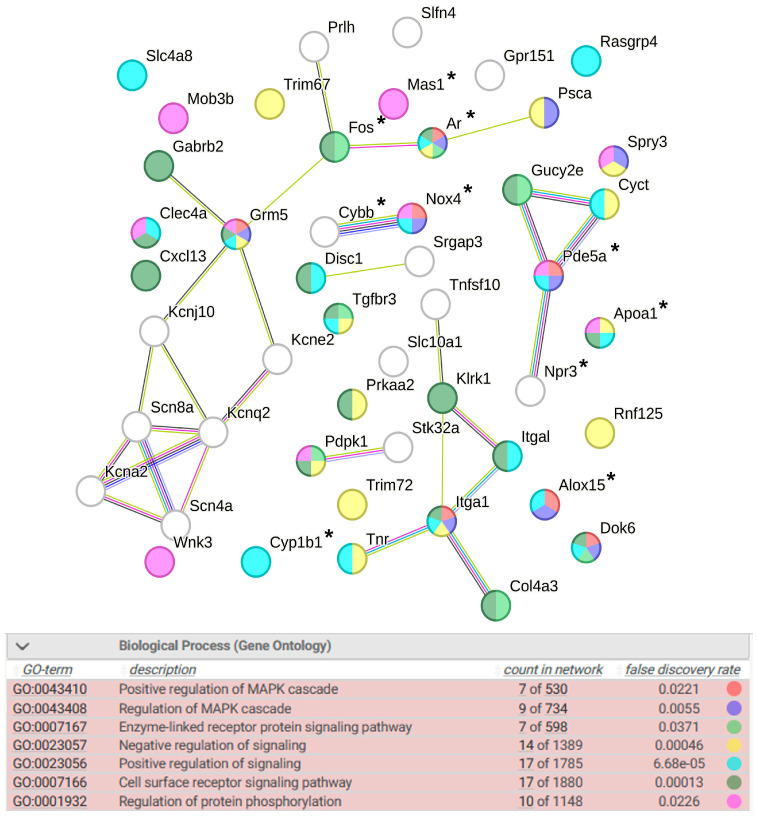
Functional annotation of 51 ISIAH strain-specific DEGs associated with signaling. Purple lines indicate experimentally determined interactions; blue lines denote known interactions from curated databases; dark blue lines represent gene co-occurrence; black lines indicate co-expression; and green lines represent the results of text mining. Protein–protein interaction (PPI) enrichment *p*-value: 9.64 × 10^−10^. *, genes associated with hypertension.

**Figure 7 ijms-25-06680-f007:**
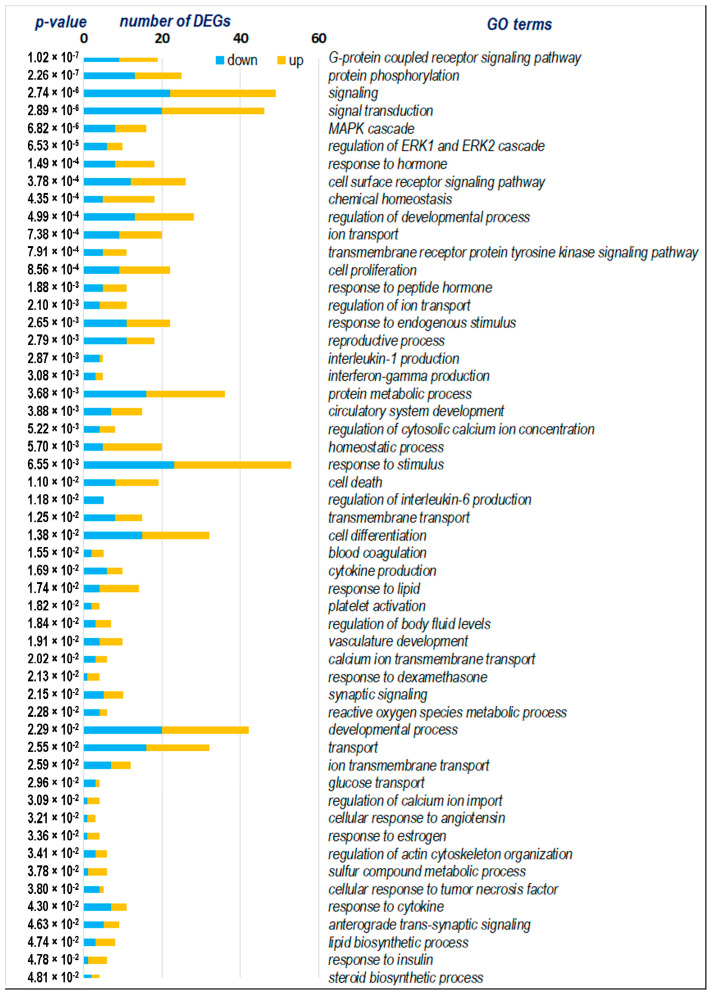
Gene ontology (GO) terms for 85 WAG strain-specific DEGs.

**Figure 11 ijms-25-06680-f011:**
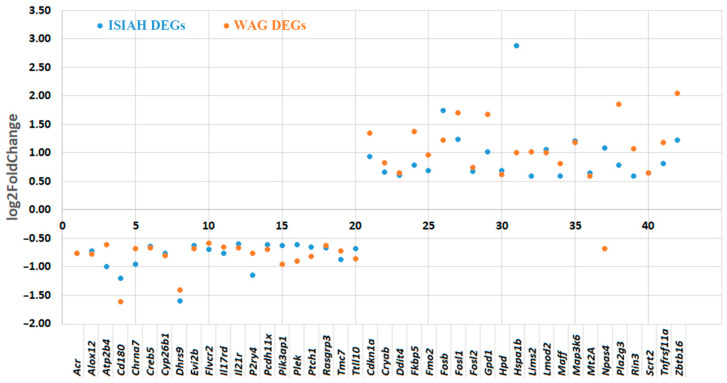
Changes in the transcription level of common 42 DEGs in the hypothalamus of ISIAH and WAG rats when exposed to single restraint stress for 2 h.

**Figure 12 ijms-25-06680-f012:**
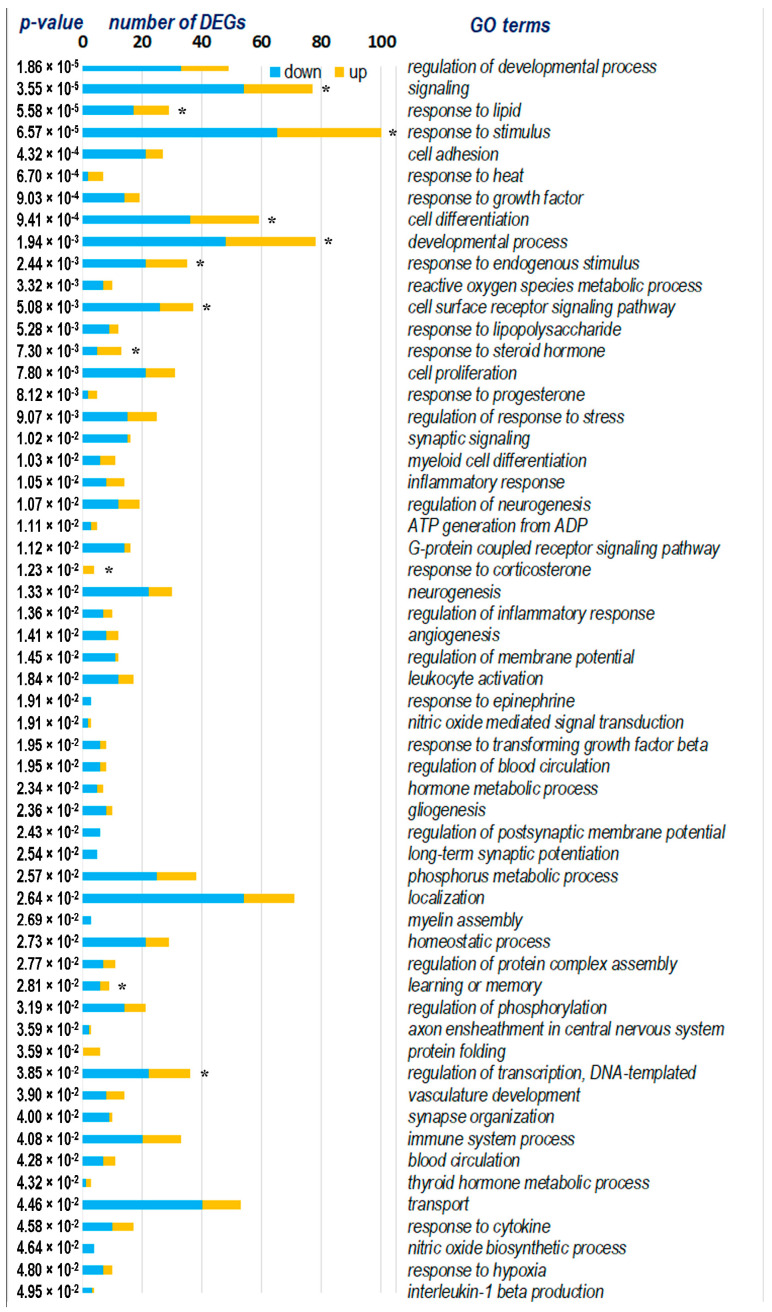
Gene ontology (GO) terms for common 144 DEGs. * the gene group includes the *Npas4* gene, which alternatively alters the level of transcription in the hypothalamus of hypertensive ISIAH and normotensive WAG rats.

**Figure 13 ijms-25-06680-f013:**
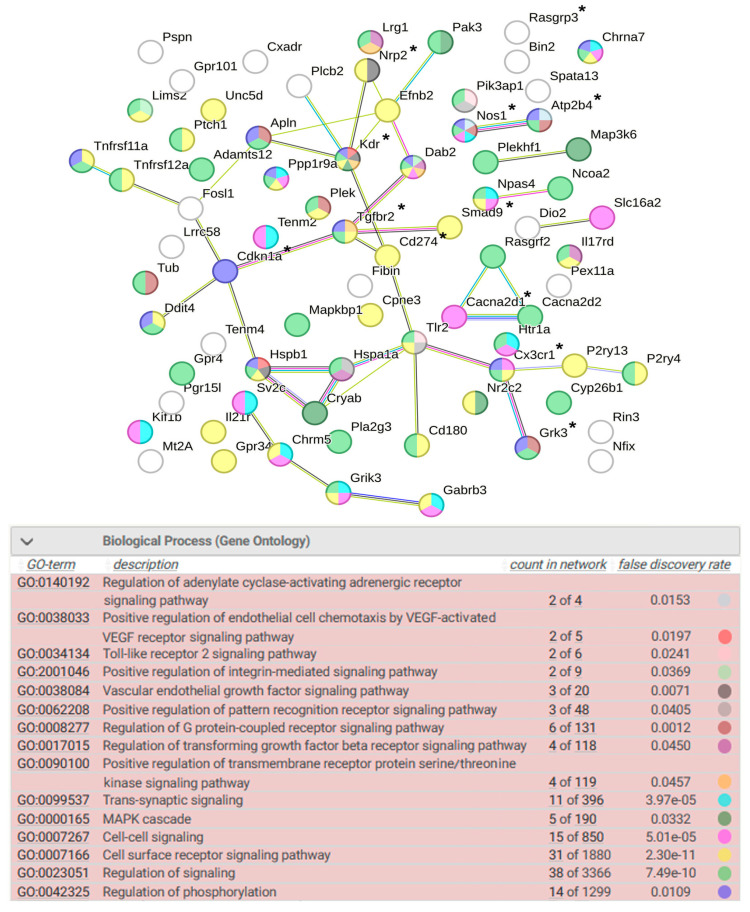
Functional annotation of 77 common DEGs associated with signaling. Purple lines indicate experimentally determined interactions; blue lines denote known interactions from curated databases; dark blue lines represent gene co-occurrence; black lines indicate co-expression; and green lines represent the results of text mining. Protein–protein interaction (PPI) enrichment *p*-value: 8.9 × 10^−7^. *, genes associated with hypertension.

**Figure 14 ijms-25-06680-f014:**
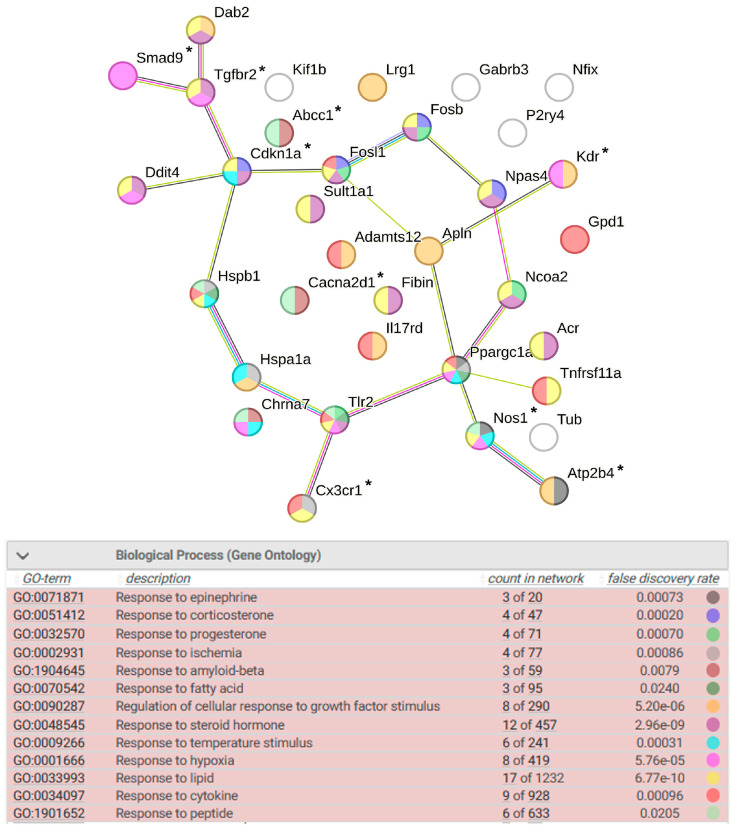
Functional annotation of 35 common DEGs associated with response to endogenous stimulus. Purple lines indicate experimentally determined interactions; blue lines denote known interactions from curated databases; dark blue lines represent gene co-occurrence; black lines indicate co-expression; and green lines represent the results of text mining. Protein–protein interaction (PPI) enrichment *p*-value: 2.01 × 10^−6^. *, genes associated with hypertension.

**Figure 15 ijms-25-06680-f015:**
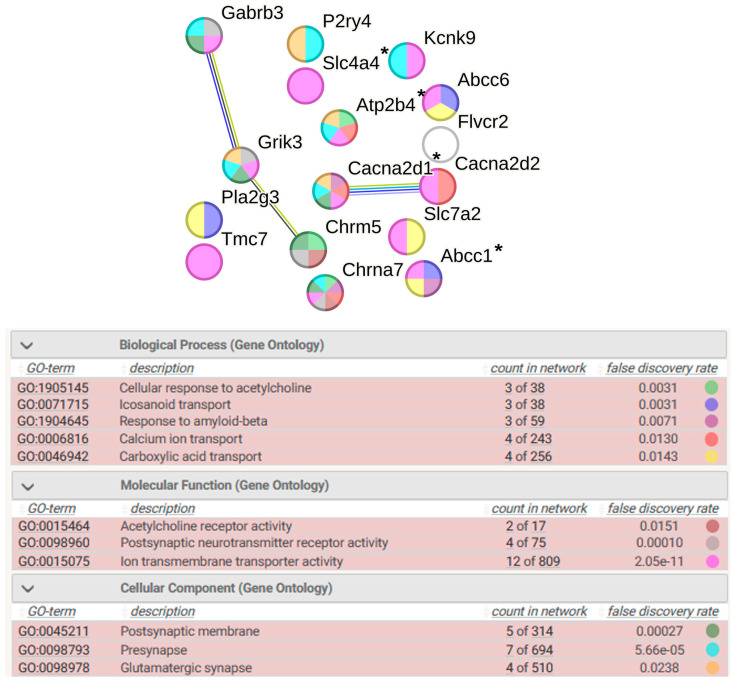
Functional annotation of 16 common DEGs associated with ion transport. Purple lines indicate experimentally determined interactions; blue lines denote known interactions from curated databases; dark blue lines represent gene co-occurrence; black lines indicate co-expression; and green lines represent the results of text mining. Protein–protein interaction (PPI) enrichment *p*-value: 0.00224. *, genes associated with hypertension.

**Figure 16 ijms-25-06680-f016:**
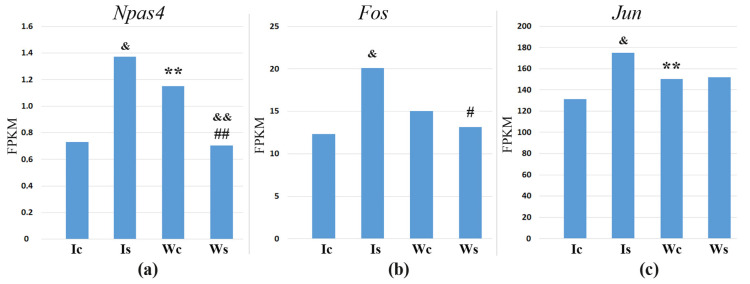
Changes in gene transcription levels in hypothalamus of ISIAH and WAG rats at rest and after exposure to restraint stress for 2 h. (**a**). *Npas4*; (**b**). *Fos*; (**c**). *Jun*; **, *p* < 0.01 (Ic_Wc comparison); #, *p* < 0.05; ##, *p* < 0.01 (Is_Ws comparison); &, *p* < 0.05; &&, *p* < 0.01 stress response (Is_Ic and Ws_Wc comparisons). Ic, ISIAH_control; Wc, WAG_control; Is, ISIAH_stress; Ws, WAG_stress.

**Figure 17 ijms-25-06680-f017:**
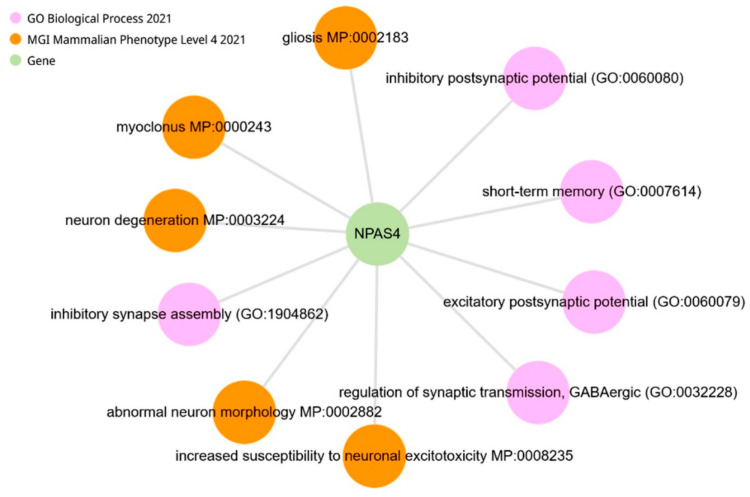
Querying NPAS4 (analyzed using the Enrichr-KG resource).

**Figure 18 ijms-25-06680-f018:**
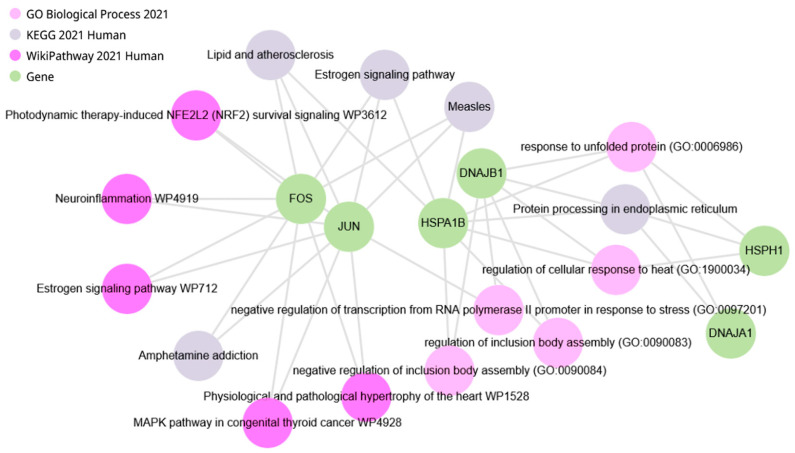
Functional analysis of ISIAH DEGs whose expression correlates with *Npas4* gene expression (analyzed using the Enrichr-KG resource).

**Figure 19 ijms-25-06680-f019:**
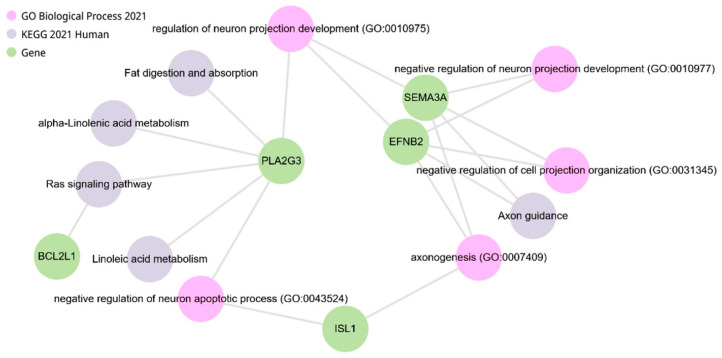
Functional analysis of WAG DEGs whose expression correlates with *Npas4* gene expression (analyzed using the Enrichr-KG resource).

**Table 1 ijms-25-06680-t001:** Transcription factor DEGs representing a specific stress response in the hypothalamus of ISIAH rats.

ISIAH DEGs	log2Fold Change	padj	GeneID	Description
*Ar* *	−0.84	6.27 × 10^−3^	24208	androgen receptor
*Arid5b*	−0.60	1.45 × 10^−4^	309728	AT-rich interaction domain 5B
*Egr4*	0.98	1.55 × 10^−3^	25129	early growth response 4
*Fos* *	0.76	2.94 × 10^−2^	314322	Fos proto-oncogene, AP-1 transcription factor subunit
*Foxn3*	−0.62	1.64 × 10^−4^	314374	forkhead box N3
*Hivep3*	−0.66	1.39 × 10^−5^	313557	HIVEP zinc finger 3
*Klf12*	−0.62	3.98 × 10^−2^	306110	Kruppel-like factor 12
*Klf13*	−0.95	1.65 × 10^−3^	499171	Kruppel-like factor 13
*Maml3*	−0.60	1.91 × 10^−2^	310405	mastermind-like transcriptional coactivator 3
*Nfe2l3*	1.12	2.50 × 10^−5^	312331	nuclear factor, erythroid 2-like 3
*Prox1*	−0.65	3.37 × 10^−4^	305066	prospero homeobox 1
*Sox6*	−0.71	1.01 × 10^−3^	293165	SRY-box transcription factor 6

* DEGs associated with hypertension.

**Table 2 ijms-25-06680-t002:** DEGs associated with arterial hypertension and representing a specific stress response in the hypothalamus of ISIAH rats.

ISIAH DEGs	log2Fold Change	padj	GeneID	Description
*Alox15*	1.24	1.98 × 10^−2^	81639	arachidonate 15-lipoxygenase
*Apoa1*	0.75	2.24 × 10^−3^	25081	apolipoprotein A1
*Ar* #	−0.84	6.27 × 10^−3^	24208	androgen receptor
*Cybb*	−0.78	1.27 × 10^−2^	66021	cytochrome b-245 beta chain
*Cyp1b1*	−0.78	3.50 × 10^−3^	25426	cytochrome P450, family 1, subfamily b, polypeptide 1
*F5*	−1.79	4.62 × 10^−2^	304929	coagulation factor V
*Fos* #	0.76	2.94 × 10^−2^	314322	Fos proto-oncogene, AP-1 transcription factor subunit
*Kcna3*	−0.66	4.83 × 10^−2^	29731	potassium voltage-gated channel subfamily A member 3
*Kcnmb1*	−0.87	3.08 × 10^−2^	29747	potassium calcium-activated channel subfamily M regulatory beta subunit 1
*Map2*	−0.71	1.13 × 10^−2^	25595	microtubule-associated protein 2
*Mas1*	−0.63	1.09 × 10^−2^	25153	MAS1 proto-oncogene, G protein-coupled receptor
*Nox4*	−0.74	2.85 × 10^−2^	85431	NADPH oxidase 4
*Npr3*	−0.97	1.25 × 10^−2^	25339	natriuretic peptide receptor 3
*Pde5a*	−0.60	4.75 × 10^−3^	171115	phosphodiesterase 5A
*Trpc5*	−0.69	6.28 × 10^−4^	140933	transient receptor potential cation channel, subfamily C, member 5

# DEGs encoding transcription factors.

**Table 3 ijms-25-06680-t003:** Upregulated DEGs characterizing the strain-specific restraint stress response in the hypothalamus of hypertensive ISIAH rats.

ISIAH DEGs	log2Fold Change	padj	Gene ID	Description
*Alox15* *	1.24	1.98 × 10^−2^	81639	arachidonate 15-lipoxygenase
*Apoa1* *	0.75	2.24 × 10^−3^	25081	apolipoprotein A1
*Egr4* #	0.98	1.55 × 10^−3^	25129	early growth response 4
*Fos* * #	0.76	2.94 × 10^−2^	314322	Fos proto-oncogene, AP-1 transcription factor subunit
*Gpr151*	2.21	2.51 × 10^−2^	307475	G protein-coupled receptor 151
*Kel*	0.88	2.82 × 10^−2^	297025	Kell metallo-endopeptidase (Kell blood group)
*Nfe2l3* #	1.12	2.50 × 10^−5^	312331	nuclear factor, erythroid 2-like 3
*Prlh*	1.03	4.65 × 10^−2^	63850	prolactin-releasing hormone
*Slc4a1*	1.23	2.55 × 10^−2^	24779	solute carrier family 4 member 1 (Diego blood group)
*Tekt4*	0.62	3.02 × 10^−9^	302991	tektin 4

* DEGs associated with hypertension; # DEGs encoding transcription factors.

**Table 4 ijms-25-06680-t004:** Upregulated DEGs associated with response to stimulus.

ISIAH DEGs	log2Fold Change	padj	GeneID	Description
*Gpr151*	2.21	2.51 × 10^−2^	307475	G protein-coupled receptor 151
*Alox15* *	1.24	1.98 × 10^−2^	81639	arachidonate 15-lipoxygenase
*Slc4a1*	1.23	2.55 × 10^−2^	24779	solute carrier family 4 member 1 (Diego blood group)
*Prlh*	1.03	4.65 × 10^−2^	63850	prolactin-releasing hormone
*Egr4* #	0.98	1.55 × 10^−3^	25129	early growth response 4
*Fos* * #	0.76	2.94 × 10^−2^	314322	Fos proto-oncogene, AP-1 transcription factor subunit
*Apoa1* *	0.75	2.24 × 10^−3^	25081	apolipoprotein A1

* DEGs associated with hypertension; # DEGs encoding transcription factors.

**Table 5 ijms-25-06680-t005:** DEGs associated with response to calcium ion in the hypothalamus of ISIAH rats.

ISIAH DEGs	log2Fold Change	padj	Description
*Alox15* *	1.24	1.98 × 10^−2^	arachidonate 15-lipoxygenase
*Fos* * #	0.76	2.94 × 10^−2^	Fos proto-oncogene, AP-1 transcription factor subunit
*Kcnmb1* *	−0.87	3.08 × 10^−2^	potassium calcium-activated channel subfamily M regulatory beta subunit 1
*Kcnq2*	−0.71	2.23 × 10^−3^	potassium voltage-gated channel subfamily Q member 2
*Prkaa2*	−0.76	1.11 × 10^−2^	protein kinase AMP-activated catalytic subunit alpha 2

* DEGs associated with hypertension; # DEGs encoding transcription factors.

**Table 6 ijms-25-06680-t006:** Transcription factor DEGs.

WAG DEGs	log2Fold Change	padj	GeneID	Description
*Esr1* *	−0.76	4.84 × 10^−3^	24890	estrogen receptor 1
*Hif3a* *	1.40	4.62 × 10^−8^	64345	hypoxia-inducible factor 3 subunit alpha
*Pax4*	−0.68	3.26 × 10^−2^	83630	paired box 4
*Pou3f4*	−0.70	5.35 × 10^−5^	29589	POU class 3 homeobox 4
*Shox2*	1.66	1.24 × 10^−4^	25546	short-stature homeobox 2
*Smyd1*	−0.80	2.88 × 10^−2^	297333	SET and MYND domain-containing 1
*Tcf7l2*	0.69	1.68 × 10^−2^	679869	transcription factor 7 like 2

* DEGs associated with hypertension.

**Table 7 ijms-25-06680-t007:** DEGs associated with arterial hypertension.

WAG DEGs	log2Fold Change	padj	GeneID	Description
*Adra1b*	0.59	8.45 × 10^−3^	24173	adrenoceptor alpha 1B
*Adra2b*	0.93	1.98 × 10^−2^	24174	adrenoceptor alpha 2B
*Esr1* #	−0.76	4.84 × 10^−3^	24890	estrogen receptor 1
*Fzd5*	−0.63	8.32 × 10^−4^	317674	frizzled class receptor 5
*Hif3a* #	1.40	4.62 × 10^−8^	64345	hypoxia-inducible factor 3 subunit alpha
*Nos2*	−0.96	4.90 × 10^−2^	24599	nitric oxide synthase 2
*Prkcd*	1.19	2.16 × 10^−2^	170538	protein kinase C, delta
*Serpine1*	0.59	2.13 × 10^−4^	24617	serpin family E member 1
*Sgk1*	1.20	7.82 × 10^−5^	29517	serum/glucocorticoid regulated kinase 1

# DEGs encoding transcription factors.

**Table 8 ijms-25-06680-t008:** Common transcription factor DEGs.

DEGs	Gene ID	ISIAH DEGs		WAG DEGs		Description
		log2Fold Change	padj	log2Fold Change	padj	
*Dab2*	79128	−0.63	7.21 × 10^−5^	−0.27	3.22 × 10^−2^	DAB adaptor protein 2
*Fosb*	100360880	1.75	3.36 × 10^−19^	1.23	1.50 × 10^−3^	FosB proto-oncogene, AP-1 transcription factor subunit
*Fosl1*	25445	1.23	7.82 × 10^−4^	1.70	2.15 × 10^−6^	FOS like 1, AP-1 transcription factor subunit
*Fosl2*	25446	0.68	2.28 × 10^−2^	0.74	3.49 × 10^−3^	FOS like 2, AP-1 transcription factor subunit
*Irf8* *	292060	−0.39	2.17 × 10^−2^	−0.61	4.24 × 10^−9^	interferon regulatory factor 8
*Maff*	366960	0.59	3.12 × 10^−4^	0.82	4.57 × 10^−9^	MAF bZIP transcription factor F
*Mlxipl*	171078	−0.46	3.77 × 10^−3^	−0.61	1.07 × 10^−6^	MLX-interacting protein-like
*Ncoa2*	83724	−0.82	2.25 × 10^−4^	−0.36	1.01 × 10^−2^	nuclear receptor coactivator 2
*Nfib*	29227	−0.67	1.46 × 10^−2^	−0.40	4.13 × 10^−2^	nuclear factor I/B
*Nfix*	81524	−0.60	4.10 × 10^−4^	−0.25	3.35 × 10^−2^	nuclear factor I X
*Nr2c2*	50659	−0.61	5.60 × 10^−3^	−0.33	2.04 × 10^−2^	nuclear receptor subfamily 2, group C, member 2
*Plag1*	297804	−0.62	1.82 × 10^−2^	−0.49	1.52 × 10^−2^	PLAG1 zinc finger
*Pou2f2*	117058	−1.00	1.56 × 10^−5^	−0.56	2.19 × 10^−3^	POU class 2 homeobox 2
*Ppargc1a*	83516	−0.63	1.31 × 10^−4^	−0.50	4.50 × 10^−4^	PPARG coactivator 1 alpha
*Scrt2*	366229	0.65	3.04 × 10^−6^	0.64	4.68 × 10^−6^	scratch family transcriptional repressor 2
*Smad9* *	85435	−0.65	2.68 × 10^−2^	−0.49	3.86 × 10^−2^	SMAD family member 9
*Tsc22d3*	83514	0.55	4.67 × 10^−13^	0.74	2.98 × 10^−37^	TSC22 domain family, member 3
*Zbtb16*	353227	1.23	9.41 × 10^−7^	2.05	2.79 × 10^−52^	zinc finger and BTB domain-containing 16

* DEGs associated with hypertension.

**Table 9 ijms-25-06680-t009:** Common DEGs associated with arterial hypertension.

DEGs	Gene ID	ISIAH DEGs		WAG DEGs		Description
		log2Fold Change	padj	log2Fold Change	padj	
*Abcc1*	24565	−0.61	2.97 × 10^−6^	−0.40	1.63 × 10^−7^	ATP-binding cassette subfamily C member 1
*Alox12*	287454	−0.72	3.07 × 10^−2^	−0.78	8.06 × 10^−3^	arachidonate 12-lipoxygenase, 12S-type
*Apln*	58812	0.47	7.24 × 10^−6^	0.71	6.87 × 10^−24^	apelin
*Atp2b4*	29600	−1.00	2.75 × 10^−3^	−0.61	2.65 × 10^−8^	ATPase plasma membrane Ca^2+^ transporting 4
*Cacna2d1*	25399	−0.66	1.83 × 10^−2^	−0.41	1.69 × 10^−4^	calcium voltage-gated channel auxiliary subunit alpha2delta 1
*Cd274*	499342	−0.66	1.51 × 10^−2^	−0.46	3.06 × 10^−2^	CD274 molecule
*Cdkn1a*	114851	0.93	2.04 × 10^−15^	1.35	4.53 × 10^−64^	cyclin-dependent kinase inhibitor 1A
*Cx3cr1*	171056	−0.70	9.57 × 10^−11^	−0.44	8.04 × 10^−11^	C-X3-C motif chemokine receptor 1
*Dio2*	65162	0.54	1.50 × 10^−4^	0.94	3.93 × 10^−19^	iodothyronine deiodinase 2
*Fndc3b*	294925	−0.60	2.44 × 10^−2^	−0.40	6.91 × 10^−3^	fibronectin type III domain-containing 3B
*Grk3*	25372	−0.73	2.80 × 10^−5^	−0.39	1.93 × 10^−4^	G protein-coupled receptor kinase 3
*Irf8* #	292060	−0.39	2.17 × 10^−2^	−0.61	4.24 × 10^−9^	interferon regulatory factor 8
*Kdr*	25589	−0.61	2.31 × 10^−13^	−0.57	4.42 × 10^−10^	kinase insert domain receptor
*Nos1*	24598	−0.82	1.59 × 10^−2^	−0.50	3.80 × 10^−4^	nitric oxide synthase 1
*Nrp2*	81527	−0.59	4.60 × 10^−4^	−0.40	5.40 × 10^−5^	neuropilin 2
*Rasgrp3*	313874	−0.66	8.28 × 10^−11^	−0.62	1.32 × 10^−17^	RAS guanyl-releasing protein 3
*Slc4a4*	84484	−0.67	4.77 × 10^−3^	−0.35	3.07 × 10^−4^	solute carrier family 4 member 4
*Smad9* #	85435	−0.65	2.68 × 10^−2^	−0.49	3.86 × 10^−2^	SMAD family member 9
*Tgfbr2*	81810	−0.67	4.75 × 10^−4^	−0.27	2.82 × 10^−2^	transforming growth factor, beta receptor 2
*Tlr2*	310553	−0.54	2.71 × 10^−5^	−0.81	1.53 × 10^−11^	toll-like receptor 2

# DEGs encoding transcription factors.

**Table 10 ijms-25-06680-t010:** Twenty top statistically significant correlations of *Npas4* gene expression with other DEGs detected in the hypothalamus of ISIAH and WAG rats.

ISIAH DEGs		WAG DEGs	
Gene_Symbol	r	Gene_Symbol	r
*Jun* * #	0.88	*Speg*	0.90
*Fos* * #	0.87	*Isl1* #	0.89
*P4ha1*	0.84	*Grifin*	−0.89
*Apex2*	0.82	*LOC361985*	−0.89
*Rnf31*	−0.81	*Ecel1*	0.87
*Ahsa2*	0.81	*Zc3h6*	0.87
*Ccdc117*	0.80	*Sema3a*	0.87
*Pcdhb21*	−0.80	*Bcl2l1* *	−0.86
*Edrf1*	0.79	*Sult5a1*	−0.86
*Hspa1b*	0.79	*Rnd2*	−0.86
*Dnajb1*	0.78	*Hnrnpr* #	0.84
*RGD1309748*	0.78	*Tes*	−0.84
*Dnaja1*	0.77	*Mlxipl* #	0.84
*Tspy26*	−0.77	*Efnb2*	0.84
*Dnaaf2*	0.77	*Prmt3*	−0.84
*Hsph1*	0.75	*Zfp189* #	−0.84
*Vps33b*	−0.75	*Mbd6*	0.84
*Tmem129*	0.74	*Pnma3*	0.83
*Zfp410* #	0.74	*Pla2g3*	−0.83
*Dvl2*	−0.74	*Cln8*	−0.83

* DEGs associated with hypertension; # DEGs encoding transcription factors.

## Data Availability

All relevant data are available in Appendix A.

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
