# Peer review of "Effect of Short-Term Restraint Stress on the Hypothalamic Transcriptome Profiles of Rats with Inherited Stress-Induced Arterial Hypertension (ISIAH) and Normotensive Wistar Albino Glaxo (WAG) Rats"

_ijms, 2024, doi:10.3390/ijms25126680_

Round 1

Reviewer 1 Report

Comments and Suggestions for Authors

The hypothalamus plays a pivotal role in modulating blood pressure and stress responses. This manuscript has studied rats with inherited stress-induced arterial hypertension (ISIAH). The rats were exposed to a 2-hour restraint stress, mimicking human emotional stress. The authors compared the hypothalamic transcriptome profiles between hypertensive ISIAH and normotensive WAG rats to understand the underlying molecular mechanisms involved in emotional stress. The results demonstrated that Fos and Jun gene transcription might be critical in neuronal activation. The hypothalamus Npas4 gene is down-regulated in the WAG rats but is up-regulated in the ISIAH rats, suggesting the strain-dependent response to stress. The topic is of significance. It would open up new strategies for developing therapeutics to care for emotional stress-induced disorders. 

I would have some comments below.

1. Rearrange the manuscript's structure. I recommend placing the Methods section before the Results section. This change will allow readers to better understand the experimental design and procedures before delving into the findings.

2. Regarding the animal models, please clarify if WAG has a genetic background similar to that of ISIAH rats.

3. In Line 354, three genes are associated with hypertension and marked with an asterisk in Figure 11. The same is true in Figure 10, etc.

4. I would suggest adding a paragraph on limit and perspective.

5. Minor revision, for example,

In line 19, in the modern human......

In line 22, revise to "......profiles of the hypothalamus......"

In line 36, add a coma "......neuronal excitability, taking into account......"

5. Typo, for example, in line 229, "Table"

Comments on the Quality of English Language

5. Minor revision, for example,

In line 19, in the modern human......

In line 22, revise to "......profiles of the hypothalamus......"

In line 36, add a coma "......neuronal excitability, taking into account......"

5. Typo, for example, in line 229, "Table"

Author Response

The authors thank the reviewer for his constructive comments, which helped to significantly improve the text of the manuscript. Corrections made to the text of the manuscript are shown in red.

The hypothalamus plays a pivotal role in modulating blood pressure and stress responses. This manuscript has studied rats with inherited stress-induced arterial hypertension (ISIAH). The rats were exposed to a 2-hour restraint stress, mimicking human emotional stress. The authors compared the hypothalamic transcriptome profiles between hypertensive ISIAH and normotensive WAG rats to understand the underlying molecular mechanisms involved in emotional stress. The results demonstrated that Fos and Jun gene transcription might be critical in neuronal activation. The hypothalamus Npas4 gene is down-regulated in the WAG rats but is up-regulated in the ISIAH rats, suggesting the strain-dependent response to stress. The topic is of significance. It would open up new strategies for developing therapeutics to care for emotional stress-induced disorders. 

I would have some comments below.

  1. Rearrange the manuscript's structure. I recommend placing the Methods section before the Results section. This change will allow readers to better understand the experimental design and procedures before delving into the findings.

Answer: Unfortunately, we cannot move the Methods section and place this information before the results, since according to the instructions for authors of the IJMS journal Research manuscripts should comprise:

  • Front matter: Title, Author list, Affiliations, Abstract, Keywords.
  • Research manuscript sections: Introduction, Results, Discussion, Materials and Methods, Conclusions (optional).
  • Back matter: Supplementary Materials, Acknowledgments, Author Contributions, Conflicts of Interest, References.
  1. Regarding the animal models, please clarify if WAG has a genetic background similar to that of ISIAH rats.

Answer: WAG rat strain was derived from outbred Wistar stock (Bacharach, Glaxo Labs., U.K., 1924) and then was selected as an inbred strain at F83 (Harrington 1964). Both the hypertensive ISIAH and normotensive WAG rat strains were derived from closely related outbred populations of Wistar Albino rats. Inbred WAG rats are characterized as normotensive strain suitable as a control group in experiments with hypertensive rats. Earlier, we have defined the WAG rats as a strain with lower behavioral, hormonal and blood pressure response to emotional stress as compared to the ISIAH rats (Markel A.L. Development of a new strain of rats with inherited stress-induced arterial hypertension. In: Genetic Hypertension. Ed. J. Sassard, Colloque INSERM, John Libbey Eurotext Ltd, 1992, vol. 218, pp.405-407. Markel A.L., Maslova L.N., Shishkina G.T., Mahanova N.A., Jacobson G.S. Developmental Influences on Blood Pressure Regulation in ISIAH Rats. In: R.McCarty, D.A.Blizard and R.L.Chevalier (Eds.) Development of the Hypertensive Phenotype: Basic and Clinical Studies. In the series Handbook of Hypertension. Amsterdam, Elsevier. 1999, 493-520).

This information is included in section 4.1. (Lines 692—696)

  1. In Line 354, three genes are associated with hypertension and marked with an asterisk in Figure 11. The same is true in Figure 10, etc.

Answer: DEGs associated with hypertension are marked in the figures.

  1. I would suggest adding a paragraph on limit and perspective.

Answer: The following text was added at the end of the discussion (Lines 673-686):

The limitations of our study include the fact that the results, while suggesting a significant role of the Npas4 gene in the regulation of the response to stress, do not allow us to associate changes in its activity with hypertension. The association of Npas4 gene expression with hypertension has not yet been established in other hypertensive animal models. However, there are studies showing that Npas4 polymorphisms contribute to the cardiovascular diseases risk (Coronary Heart Disease) [57]. Npas4 has also been shown to increase transcription levels in the brain under conditions of dehydration [58]. It is known that dehydration has a profound influence on neuroexcitability, but on the other hand, it is also known that water deprivation can change the water-salt balance of the body. An increase in the concentration of sodium ions in body fluids activates the sympathetic neural activity leading to hypertension [59]. Thus, it can be hypothesized that Npas4 may be involved in the regulation of BP. This aspect may be very promising for further study of the role of Npas4 not only as a factor influencing neuroexcitability, but also as an important link in the formation of hypertensive status.

  1. Minor revision, for example,

Answer:  All done. Thank you

In line 19, in the modern human......

In line 22, revise to "......profiles of the hypothalamus......"

In line 36, add a coma "......neuronal excitability, taking into account......"

Typo, for example, in line 229, "Table"

Comments on the Quality of English Language

  1. Minor revision, for example,

In line 19, in the modern human......

In line 22, revise to "......profiles of the hypothalamus......"

In line 36, add a coma "......neuronal excitability, taking into account......"

  1. Typo, for example, in line 229, "Table"

Reviewer 2 Report

Comments and Suggestions for Authors

Oshchepkov and co-workers used hypertensive ISIAH and normotensive WAG rats exposed to restraint stress. They analyzed differential gene expression in the hypothalamus with subsequent bioinformatical analysis.

The authors present an extremely long manuscript of 37 pages, containing 22 figures and 10 tables... this abundance of data makes the manuscript very confusing, you lose the overview after just a few pages. The model used and the results of the analyses are certainly interesting and worth publishing. But please limit yourself to the essentials and move all other information to the supplement.

Comments:

- Abstract, line 23: Please specify restraint stress more precisely.

- The somewhat cryptic labels for the experimental conditions should be changed.

- Tables: In all tables, the commas must be replaced by full stops.

- e.g. Figure 3: Please specify the values gives on top of the bars.

- Line 229: Translate to English.

- Line 580: Such figures do not belong in the discussion, but in the results section.

Comments on the Quality of English Language

Author Response

The authors thank the reviewer for his constructive comments, which helped to significantly improve the text of the manuscript. Corrections made to the text of the manuscript are shown in red.

Oshchepkov and co-workers used hypertensive ISIAH and normotensive WAG rats exposed to restraint stress. They analyzed differential gene expression in the hypothalamus with subsequent bioinformatical analysis.

The authors present an extremely long manuscript of 37 pages, containing 22 figures and 10 tables... this abundance of data makes the manuscript very confusing, you lose the overview after just a few pages. The model used and the results of the analyses are certainly interesting and worth publishing.

But please limit yourself to the essentials and move all other information to the supplement.

Answer: Several drawings were reconstructed to reduce their size; three drawings were moved to Supplementary materials. Also we have adjusted Figure 2, bringing it and the corresponding description in the text into line with the lists of genes under consideration. The volume of the manuscript was reduced by 5 pages.

Comments:

- Abstract, line 23: Please specify restraint stress more precisely.

Answer:  Done. (Lines 23-24)

- The somewhat cryptic labels for the experimental conditions should be changed.

Answer: The labels Io_Ik and Wo_Wk are changed to ISIAH_DEGs for ISIAH_stress_ISIAH_control and WAG_DEGs for WAG_stress_WAG_control (Lines 747-749)

- Tables: In all tables, the commas must be replaced by full stops.

Answer: - Done

- e.g. Figure 3: Please specify the values gives on top of the bars.

Answer: - Done

- Line 229: Translate to English.

Answer: - Done

- Line 580: Such figures do not belong in the discussion, but in the results section.

Answer: – Figure 22 moved to results section (Lines 452-463)

Reviewer 3 Report

Comments and Suggestions for Authors

Oshchepkov et al. performed in their interesting and modern  study a comparative analysis of transcriptome response to a single short-term (2 hours) restraint stress in the hypothalamus of Inherited Stress-Induced Arterial Hypertension (ISIAH) and their normotensive control Wistar albino Glaxo (WAG) rats.

In my opinion Authors should discuss in the additional subchapter “Limitations” whether their results are specific to the experimental conditions described in the current publication or  if they are more general i.e. applicable to other models of hypertension [especially to spontaneously hypertensive rats (SHR), the most popular model of primary hypertension] or the type of sustained stress.

Authors state that they used a single short-term (2 hours) restraint stress. It is not entirely true, as 30 min prior to the end of the stress procedure, the cage with the rat was placed on a warm (37°C) platform in order to prepare the rat for prepare blood pressure measurement by the tail-cuff. Please discuss the impact of higher temperature on restraint stress and blood pressure (BP) values before and after stress. Moreover, two different conditions of BP measurement (in ether anesthesia and without anesthesia before and after stress, respectively) should be mentioned in Legend to Fig.1

Minor remarks:

1.     Abbreviations should be explained carefully throughout the texts and in all legends. Moreover, ISIAH and WAG should be replaced with their full names in the title.

2.     Legends to Figures should be corrected since they are not self-describing.

3.     Figure 8 is not mentioned in the text of the Results.

4.     The Introduction is too long, e.g. the second paragraph concerns basic of physiology

5.     The old literature and the literature in Russian should be replaced.

6.     How can Authors explain that resistant stress failed to modify BP in WAG?

Author Response

The authors thank the reviewer for his constructive comments, which helped to significantly improve the text of the manuscript. Corrections made to the text of the manuscript are shown in red.

Comments and Suggestions for Authors

Oshchepkov et al. performed in their interesting and modern study a comparative analysis of transcriptome response to a single short-term (2 hours) restraint stress in the hypothalamus of Inherited Stress-Induced Arterial Hypertension (ISIAH) and their normotensive control Wistar albino Glaxo (WAG) rats.

In my opinion Authors should discuss in the additional subchapter “Limitations” whether their results are specific to the experimental conditions described in the current publication or  if they are more general i.e. applicable to other models of hypertension [especially to spontaneously hypertensive rats (SHR), the most popular model of primary hypertension] or the type of sustained stress.

Answer: The following text was added at the end of the discussion (Lines 673-686):

The limitations of our study include the fact that the results, while suggesting a significant role of the Npas4 gene in the regulation of the response to stress, do not allow us to associate changes in its activity with hypertension. The association of Npas4 gene expression with hypertension has not yet been established in other hypertensive animal models. However, there are studies showing that Npas4 polymorphisms contribute to the cardiovascular diseases risk (Coronary Heart Disease) [57]. Npas4 has also been shown to increase transcription levels in the brain under conditions of dehydration [58]. It is known that dehydration has a profound influence on neuroexcitability, but on the other hand, it is also known that water deprivation can change the water-salt balance of the body. An increase in the concentration of sodium ions in body fluids activates the sympathetic neural activity leading to hypertension [59]. Thus, it can be hypothesized that Npas4 may be involved in the regulation of BP. This aspect may be very promising for further study of the role of Npas4 not only as a factor influencing neuroexcitability, but also as an important link in the formation of hypertensive status.

Authors state that they used a single short-term (2 hours) restraint stress. It is not entirely true, as 30 min prior to the end of the stress procedure, the cage with the rat was placed on a warm (37°C) platform in order to prepare the rat for prepare blood pressure measurement by the tail-cuff. Please discuss the impact of higher temperature on restraint stress and blood pressure (BP) values before and after stress.

Answer: The following text was added to the Materials and Methods section:

Thirty min before the end of the stress procedure, the cage with the rat was placed on a warm (37°C) platform to prepare the rat for blood pressure measurement. This condition of BP measurement is caused by the need to increase the pulsation of the tail artery and is accepted as a standard approach to obtain correct BP results. There were no signs of additional excitement or other negative behavioral reactions in the rats after a slight warming of the environment. (Lines 711-714).

Moreover, two different conditions of BP measurement (in ether anesthesia and without anesthesia before and after stress, respectively) should be mentioned in Legend to Fig.1-

Answer: Done (Lines 101-102).

Minor remarks:

  1. Abbreviations should be explained carefully throughout the texts and in all legends.

Answer: – Done (see lines 63, 163, 243, 282, 375, 551)

  1. Moreover, ISIAH and WAG should be replaced with their full names in the title.

Answer: – Done

  1. Legends to Figures should be corrected since they are not self-describing.

Answer: - The necessary corrections have been made to the legends of the figures.

  1. Figure 8 is not mentioned in the text of the Results.

Answer:  - please, see line 277

  1. The Introduction is too long, e.g. the second paragraph concerns basic of physiology

Answer: the second paragraph of the "Introduction" section has been shortened

  1. The old literature and the literature in Russian should be replaced.

Answer: Unfortunately, we cannot delete or replace references 21-23, since these publications describe the basic physiological characteristics of the hypertensive ISIAH rat strain that was created by Dr. A.L. Markel. But, following the reviewer's recommendation, in the discussion section we replaced several of the oldest references with newer ones - references 29, 42, 49. The literature in Russian is replaced (reference 26).

  1. How can Authors explain that resistant stress failed to modify BP in WAG?

Answer: We have added clarifications that briefly explains the possible reasons for the differences in blood pressure changes under stress in ISIAH and WAG rats (Discussion section (Lines 523-533) and Conclusions section (Lines 794–797)).

Round 2

Reviewer 2 Report

Comments and Suggestions for Authors

The manuscript still contains an unusually large number of figures and tables, but my comments have been fully implemented.

Reviewer 3 Report

Comments and Suggestions for Authors

The manuscript has been improved satisfactorily.